# A 30-meter resolution dataset of China's urban impervious surface area and green space fractions, 2000–2018

Wenhui Kuang [1], Shu Zhang[1,2], Xiaoyong Li[2,3], Dengsheng Lu[4,5]

[1]Key Laboratory of Land Surface Pattern and Simulation, Institute of Geographic Sciences and Natural Resources Research,
Chinese Academy of Sciences, Beijing 100101, China.

[2]College of Resources and Environment, University of Chinese Academy of Sciences, Beijing 10049, China.

[3]State Key Laboratory of Urban and Regional Ecology, Research Center for Eco-
environmental Sciences, Chinese Academy of Sciences, Beijing 100085, China.

[4]School of Geographical Sciences, Fujian Normal University, Fuzhou 350007, China.

[5]Fujian Provincial Key Laboratory for Subtropical Resources and Environment, Fujian Normal University, Fuzhou 350007, China

*Correspondence to*: Wenhui Kuang (kuangwh@igsnrr.ac.cn)

**Abstract.** Accurate and timely maps of urban underlying land properties at the national scale are of significance in improving habitat environment and achieving sustainable development goals. Urban impervious surface (UIS) and urban green space (UGS) are two core components for characterizing urban underlying environments. However, the UIS and UGS are often mosaicked in the urban landscape with complex structures and composites. The 'hard classification' or binary single type cannot be used effectively to delineate spatially explicit urban land surface property. Although six mainstream datasets on global or national urban land use/cover products with 30-m spatial resolution have been developed, they only provide the binary pattern or dynamic of a single urban land type, which cannot effectively delineate the quantitative components or structure of intra-urban land cover. Here we proposed a new mapping strategy to acquire the multitemporal and fractional information of the essential urban land cover types at national scale through synergizing the advantage of both big data processing and human interpretation in aid of geoknowledge. Firstly, the vector polygons of urban boundaries in 2000, 2005, 2010, 2015 and 2018 were extracted from China's Land Use/cover Dataset (CLUD) derived from Landsat images. Secondly, the national settlement and vegetation percentages were retrieved using sub-pixel decomposition method through random forest algorithm using Google Earth Engine (GEE) platform. Finally, the products of China's UIS and UGS fractions (CLUD-Urban) at 30-meter resolution were developed in 2000, 2005, 2010, 2015 and 2018. We also compared our products with existing six mainstream datasets in quality and accuracy. The assessment results showed that the CLUD-Urban product has higher accuracies in urban boundaries and urban expansion detection than other products, in addition that the accurate UIS and UGS fractions were developed in each period. The overall accuracy of urban boundaries in 2000-2018 are over 92.65%; and the correlation coefficient (R) and root mean square errors (RMSE) of UIS and UGS fractions are 0.91 and 0.10, and 0.89 and 0.11, respectively. Our result indicates that the 71% pixels of urban land were mosaicked by the UIS and UGS within cities in 2018, which single UIS classification may highly increase the mapping uncertainty. The high spatial heterogeneity of urban underlying covers was exhibited with average fractions of 68.21% for UIS and 22.30% for UGS in 2018 at national scale. The

UIS and UGS increased unprecedentedly with annual rates of 1,605.56 km$^2$/yr and 627.78 km$^2$/yr in 2000-2018, driven by fast
urbanization. The CLUD-Urban mapping can fill the knowledge gap in understanding impacts of the UIS and UGS patterns
on ecosystem services and habitat environments, and is valuable for detecting the hotspots of waterlog and improving urban
greening    for    planning    and    management    practices.    The    datasets    can    be    downloaded    from
https://doi.org/10.5281/zenodo.4034161 (Kuang et al., 2020a).

## 1 Introduction

The effects of rapid urbanization on environments have been witnessed around the world (Seto et al., 2012; Bai et al.,
2018; Kuang et al., 2020b) and profoundly contribute to the changes in biosphere, hydrosphere and atmosphere (J. Wu et al.,
2014; Kuang et al., 2018). In China, a rapid urbanization process appeared in the 21$^{st}$ century (Xu and Min, 2013; Ma et al.,
2014; Bai et al., 2014; Kuang, 2012; Kuang et al., 2016),resulting in rapid increase in urban impervious surface area (UIS)
(Kuang et al., 2013; Kuang & Dou, 2020; Lu et al., 2008). This process further triggered various urban environmental problems
such as urban heat island and urban flooding (Haase et al., 2014; Hamdi & Schayes; 2007; Kuang, 2011; Kuang et al., 2015;
Kuang et al., 2017; Xu, 2006; Zhang et al., 2017). Although many green areas were constructed in Chinese cities recently,
China has relatively lower urban green space (UGS) percentage than developed countries such as United States (Nowak and
Greenfield, 2012; Kuang et al., 2014). These urban environmental problems triggered the urgency of developing accurate
urban land-cover datasets with high spatial resolution for delineating the underlying urban environments. Along with the
development of earth observation technologies, remote sensing has become the mainstream method for mapping UIS and UGS,
and monitoring their changes (Weng, 2012; Wang et al., 2013; Lu et al., 2014; Lu et al., 2018; Zhang et al., 2009).

Various land-use products such as the Global Land Cover product (GlobeLand30) (Chen et al., 2015), the University of
Maryland (UMD) Land Cover Classification (Hansen et al., 2000), Moderate Resolution Imaging Spectroradiometer
(MODIS)-based land use/cover products (Friedl et al., 2010), GlobCover (Bontemps et al., 2011) and finer resolution
observation and monitoring of global land cover (FROM-GLC) (Gong et al., 2013) are freely available worldwide (Grekousis
et al., 2015; Dong et al., 2018). These products have different definitions of urban areas or settlements due to their different
classification systems, such as the International Geosphere-Biosphere Programme (IGBP) (Belward, 1996). Some urban land
dataset, such as Normalized Urban Areas Composite Index (NUACI), which were constructed by supervised learning
approaches were released at national or global scale with spatial resolutions from 30 m to 1 km (Liu et al., 2018; He et al.,
2019; Gong et al., 2019). Others such as built-up grid of the Global Human Settlement Layer (GHS Built) (Pesaresi et al.,
2013) and Global Urban Footprint (GUF) (Esch et al., 2017, 2018) have been published too. Most urban land products focused
on built-up land or urban area classification but cannot delineate urban land as a heterogeneous unit consisting of UIS, UGS
and others (Chen et al., 2015). Therefore, few urban land products provided intra-urban UIS and UGS fractions at the sub-
pixel level.

Detailed UIS dataset inside a city is required as a primary urban environmental index. Numerous studies on impervious surface mapping at the national scale mainly rely on medium-low spatial resolution remotely sensed data such as MODIS and Defense Meteorological Satellite Program's Operational Linescan System (DMSP-OLS) (Gong et al., 2013; Zhou et al., 2014; Grekousis et al., 2015; Zhou et al., 2015; Kuang et al., 2016; Zhou et al., 2018). Recently, more research is shifted to employ medium-high spatial resolution data (e.g., Landsat) to improve the products (Li et al., 2018; Liu et al., 2018; Gong et al., 2019;

Gong et al., 2020; Li et al., 2020; Lin et al., 2020). The U.S. Geological Survey developed the National Land Cover Database (NLCD) and provided impervious surface fraction, percent tree canopy, land-cover classes and their changes with a spatial resolution of 30 m (Falcone and Homer, 2012; Yang et al., 2018). However, detailed intra-urban UIS and UGS dataset with 30 m spatial resolution for China at the national scale is not available yet, making it difficult to conduct detailed analysis of such applications as urban living environments.

A systematic assessment on urban land mapping algorithms indicates that previous research mainly classified urban land into a single type with 'urban area' or impervious surface area (ISA), which limits the recognition on urban environment (Reba & Seto, 2020). There are two critical challenges in mapping urban land cover composites. Firstly, the conceptual definition of urban land or ISA in previous research is unclear, thus, the spatial extent is inconsistent, resulting in large divergence in statistical area of urban land. Meanwhile, the segmentation on urban-rural boundaries was not accurate from moderate

resolution satellite images using computer-based automatic classification owing to the differences in geographic conditions, social economic conditions and land policies. Therefore, accurate mapping of urban-rural boundaries is pivotal in detecting urban land-cover change. Secondly, the spatial heterogeneity of urban surface property resulted in difficulty in decomposing urban land-cover types with complex surface materials at pixel scale, which was limited by huge amounts of data processing and storage capacities with 30-m resolution.

In reality, the urban land-cover is composed of UIS, UGS and others. UIS refers to the urban impervious surface features caused by artificial land-use activities, like building roofs, asphalt or cement roads, and parking lots. UGS is an important component of the green infrastructure of cities and provides a range of ecosystem services, including parks, trees and grass. Previous studies have proven that spectral mixture analysis (SMA) provides an effective tool to retrieve the UIS and UGS fractions from Landsat multispectral imagery (Lu and Weng, 2004, 2006; Peng et al. 2016; Kuang et al., 2018). However, this

method needs local knowledge for problem-specific analysis such as intra-urban land-cover analysis of a single city or a single urban agglomeration (Zhang and Weng, 2016; Xu et al., 2018). Although the globally standardized SMA can effectively extract substrate, dark and vegetation (Small, 2013), the UIS cannot be accurately and directly extracted from multispectral image without post-processing considering its widely spectral variation and different meanings between UIS and substrate (Lu et al., 2014). Because of the high correlation between UIS and vegetation indices in the urban landscape (Weng et al., 2004),

fractional UIS dataset can be estimated from vegetation indices using regression-based approach (Sexton et al., 2013; Wang et al., 2017).

       To address above issues, we proposed a synthetical strategy to utilize the advantage of both accurate urban boundaries' information from China's Land Use/cover Dataset (CLUD) extracted by human-computer digitalization and the retrieval of

UIS and UGS fractions through the big-data processing from GEE platform. Based on the strategy, we developed the product of national UIS and UGS fractions dataset at 30-m spatial resolution in 2000, 2005, 2010, 2015 and 2018 across China. This dataset provides foundation for urban dwellers' environments and enhance our understanding on the impacts of urbanization on ecological services and functions, and is also helpful in future researches and practices on urban planning and urban environmental sustainability.

## 2 The strategy of developing CLUD-Urban product

To acquire the accurate CLUD-Urban product, three steps were generally implemented according to our mapping strategy. Firstly, national urban boundaries in 2000-2018 were extracted from CLUD which was generated using the uniform technological flow and classification system in human-computer digitalization environment. Time series of urban boundaries and their expansions have good performance in accuracy and data quality. The national urban vector boundaries in 2000, 2005, 2010, 2015 and 2018 were converted to raster data with 30-m resolution for further processing (Fig. 1). Secondly, the settlement and vegetation fractions with 30-m resolution were retrieved using random forest algorithm in GEE platform. Thirdly, the UIS and UGS fractions with 30 m resolution were mapped through overlaying the urban boundaries of CLUD with settlement and vegetation fractions, respectively (Fig. 1). The accuracy assessment of both urban boundaries, and UIS and UGS fractions was implemented using samples from Google Earth images. The quality control was conducted throughout the data processing in mapping the CLUD-urban product. The detail description was addressed in the following sections.

[Insert Figure 1 here]

## 3 Data sources and pre-processing

Landsat is the longest-running satellite series for Earth observation. Landsat Thematic Mapper (TM), Enhanced Thematic Mapper Plus (ETM+) and Operational Land Imager (OLI) data with path ranges of 118–149 and row ranges of 23–43 in China were selected (Table 1). In mapping CLUD, Landsat TM, ETM+ and OLI in each period, China-Brazil Earth Resources Satellite (CBERS) and Huan Jing (HJ-1A/B) satellite images in 2010 were used to generate the false-colour composite images with near-infrared, red, and green spectral bands as red, green, and blue. The image enhancement was processed to improve the visual interpretation quality. The image to image registration was conducted to control the image rectification errors of less than 2 pixels (60 m). CBERS-1 and Huan Jing (HJ-1A/B) satellite images were only used in extracting the vector polygons of CLUD in 2010, which was conducted using the uniform data processing with Landsat images.

In retrieval of settlement and vegetation fractions, Landsat TM, ETM+, and OLI in each period from January to December were collected using GEE platform. SRTM Digital Elevation model data and NDVI with 30 m resolution were acquired as

input parameters to retrieve settlement and vegetation fractions. Google Earth images in selected cities with 0.6 m resolution were used to assess the accuracy of CLUD-Urban product.


[Insert Table 1 here]

## 4 Extraction of urban boundaries from CLUD

4.1 The classification system and interpretation symbols

CLUD with 30-m resolution was developed by the Chinese Academy of Sciences and has been updated from 2000 to
2018 every five or three years. This dataset can delineate land use or land cover change associated with human activities, including urbanization at a scale of 1:100,000 (Liu, Liu, Tian et al., 2005; Liu, Liu, Zhuang et al., 2005; Liu et al., 2010). This product adopted a hierarchical classification system covering the first-level six classes and the second-level twenty-five classes. Here the first-level six classes include cropland, woodland, grassland, water body, construction land, and unused land. The detailed description of each class can be found in previous publications (Liu, Liu, Zhuang et al., 2005; Zhang et al., 2014). The
construction land was divided into three second-level classes, including urban land, rural settlements, and industrial and mining lands beyond cities. Urban land was defined as a built-up area of the concentrated construction, i.e. buildings, roads, squares, green infrastructure and other lands for providing the living, industrial production, and ecosystem services for the dwellers of cities or towns. According to this definition, urban land can be megacities (more than 10 million population), megalopolis (5-10 million population), large cities (1-5 million population), medium cities (0.5-1 million population), small cities (0.2-0.5
million population), and towns (less than 0.2 million population) (Kuang, 2020a). The industrial and traffic lands outside cities are excluded in the urban land.  Based on the designed classification system, the interpretation symbols from the second-level classes were built for the false-colour composite images as a reference to aid the human-computer interpretation (Fig. 2) (Zhang et al., 2014).

[Insert Figure 2 here]

4.2 Land use and dynamic polygon interpretation

According to CLUD classification system, the vector polygons of land use classes in 2000 were digitalized through overlying the false-colour composite images in aid of interpretation symbols and the geoknowledge from each zone (Fig. 3). The polygons of urban lands were identified through using the detailed image interpretation symbols for each second-level
land use class based on Landsat or similar resolution images. Usually, the polygons of urban lands exhibit larger sizes than rural settlements and others (e.g., industrial and traffic lands) in cinerous colour ornamenting with white. The digitalized personnel differentiated the urban land from rural settlements and others based on the interpretation symbols and geo-knowledge from field investigation (Fig. 2). In the digitalization environment, each vector polygon was assigned with a code

of the second-level classes. The vector polygons of land use classes in 2000 were double checked to ensure the correct type in interpretation. The dynamic polygons were extracted through comparing the difference of two-date images and assigned the codes including the types before and after changes (Fig. 3). The land use changes within five or three years were detected using the uniform method. Finally, the land use maps in 2000, 2005, 2010, 2015 and 2018 and their changes at five- or three-year interval were generated for CLUD. The detailed technological flow can be found in previous publications (Liu, Liu, Zhuang et al., 2005; Zhang et al., 2014). An example of land use map in 2010 in Conghua district of Guangzhou city and their dynamic changes in 2010-2015 is illustrated in Fig. 3.

[Insert Figure 3 here]

4.3 Retrieval of multitemporal urban boundaries

The vector boundaries of urban extents were extracted from the CLUD land use maps in each period (Kuang et al., 2016). We also examined 10,732 urban vector polygons in 2000. The number of polygons increase to 50,061 in 2018. The urban vector boundaries were acquired from Landsat images or similar resolution images. The vector polygons of urban boundaries were converted to raster data with 30 m ✕ 30 m cell size. The dataset on urban land across China in 2000, 2005, 2010, 2015 and 2018 were generated with 30-m resolution. Here we showed urban boudaries and expansion process with 30-m resolution in cities of Xi'an, Wuhan, Guangzhou and Urumqi (Fig. 4).

[Insert Figure 4 here]

**5 Mapping UIS and UGS fractions using GEE platform**

5.1 Collection of training samples

The training samples of UIS and UGS fractions are a pivotal input parameter in random forest model for mapping national settlement and vegetation fraction. In light of large discrepancies among UIS and UGS composites in different climate zones with various geographical and social economic conditions, we collected a total of 2,570 samples from randomly selected cities in different climate zones (Schneider et al. 2010) (Fig. 5). Here we also refer to the existing UIS dataset to acquire samples with 10% intervals of the ISA fraction, and those samples primarily distributed in the homogeneous UIS or UGS areas, which might provide more effective samples and decrease the impact of imagery mismatch. The samples of UIS and UGS covered with diversified types, including buildings, roads and squares, and grass, trees from parks, road and residential green spaces. The UIS and UGS percentages were interpreted within each sample using Google Earth images (Fig. 5b1-b4). Finally, the training samples in 2000, 2005, 2010, 2015 and 2018 were used for training the random forest model, respectively.

[Insert Figure 5 here]

5.2 Retrieval of settlement and vegetation fractions using random forest model

Many previous studies have indicated that random forest is more effective and accurate in classifying urban land types than other machine learning approaches such as support vector machine (SVM) and artificial neural network (ANN) (Zhang et al., 2020). Random forest exhibits a strong capacity in processing high-dimensional datasets and has been successfully
applied to mapping global ISA at 30-m resolution (Zhang et al., 2020). In this research, we proposed a strategy to acquire the settlement and vegetation percentage at pixel scale using the advantage of random forest and big-data processing based on GEE platform.

According to sixteen global urban ecoregions based on temperature, precipitation, topographic conditions and social economic factors (Schneider et al. 2010), China has three urban ecoregions. In each urban ecoregion, the annual maximum
NDVI, and spectral bands in Landsat TM/ETM+/OLI, and the slope index derived from SRTM DEM with 30-m resolution were selected as the input parameters to run random forest model. The Landsat images were from January 1 to December 31 of each baseline year. The annual maximum NDVI ($NDVI_{max}$) was retrieved using equation (1):

$$NDVI_{max} = \mathrm{ma\,x}(NDVI_1, NDVI_2, \cdots, NDVI_i) \tag{1}$$

where $NDVI_i$ is the NDVI value of the $i^{th}$ image. Individual NDVI was calculated from Landsat images in the period between
January 1 to December 31 and all images were collected using GEE (Gorelick et al., 2017).

In GEE platform, the settlement and vegetation fractions were calculated for each urban ecoregion through using the training parametrizations. The lawn, forest or their mosaicked areas were selected as input samples in mapping UGS. A post-processing was implemented to remove the pixels with NDVI values of greater than 0.5 or DEM slope values of greater than 15º. In arid and semi-arid areas, the enhanced bare soil index (EBSI) was utilized to separate UIS from bare soils (As-syakur
et al., 2012; Li et al., 2019). As a result, the settlement and vegetation fractions with 30 m ✕ 30 m in 2000, 2005, 2010, 2015 and 2018 were generated for developing CLUD-Urban product (Fig. 6).

[Insert Figure 6 here]

5.3 Mapping of UIS and UGS fractions

The settlement and vegetation fractions with 1º ✕ 1º grid of each period were downloaded from GEE platform. In ARCGIS 10.0 software, the settlement and vegetation layers were merged respectively at provincial scale with 30 m ✕ 30 m. The national UIS and UGS fractions with 30 m ✕ 30 m resolution in 2000, 2005, 2010, 2015 and 2018 were produced through overlaying the urban boundaries of CLUD with settlement and vegetation fractions, respectively (Fig. 7, Fig. 8 and Fig. 9).

[Insert Figure 7 here]

                                              [Insert Figure 8 here]

                                              [Insert Figure 9 here]

**6 Accuracy assessment of CLUD-urban product**

        The national urban boundaries and UIS and UGS fractions were assessed through qualitative and quantitative indexes,

respectively. Firstly, we referred on the accuracy of CLUD in 2000, 2005 and 2010 from our previous publications (Liu et al.,

2010; Liu et al., 2014; Zhang et al., 2014). The accuracy of the first-level six classes – cropland, forest, grassland, built-up

area, water body and unused and of the second-level land use/cove types, including urban land, rural settlements, industrial

and traffic lands was assessed using the field investigation data and the Google Earth images (Liu et al., 2010; Liu et al., 2014;

Zhang et al., 2014). We also implemented accuracy assessment on urban boundaries of CLUD from 2000 to 2018 using overall

accuracy, producer's accuracy, and user's accuracy (Fig. 10) (Kuang et al., 2016; Kuang, 2020a).

        The validation samples for assessing the accuracy of UIS and UGS fractions were collected within urban boundaries using

a stratified random sampling method with the ISA fraction at 10% intervals. Those samples with a window size of 3×3 pixels

(90 m×90 m) were used to digitalize the UIS and UGS polygons through the human-computer interaction based on Google

Earth images (Kuang et al., 2014; Kuang, 2020b). A total of 1,869 validation samples were randomly acquired in different

regions in China and 1070 were located in changed UIS and UGS areas during 2000-2018 (Fig. 10). Mean UIS and UGS

fractions in each grid were calculated. The comparison between estimated values and validation values was conducted using

the correlation coefficient (R) and root mean square error (RMSE) (Kuang et al., 2014; Kuang, 2020).

[Insert Figure 10 here]

**7 Results**

7.1 The accuracy of CLUD-urban

        The quality check and data integration were performed for the years of 2000, 2005, 2010, 2015 and 2018 to ensure the

quality and consistency of the interpretation results. Our assessment results indicated the overall accuracy of of the first-level

land use/cover types is 98.04% in 2000, 94.3% in 2010, 91.64% in 2015, and 91.12% in 2018 (Liu et al., 2014; Zhang et al., 2014; Kuang et al., 2016; Ning et al., 2019). The built-up area has the highest accuracy among the six land use types owing to their clear urban boundaries, and the accuracy reached 98.92% in 2000 and 97.01% in 2005 according to previous assessment (Zhang et al., 2014). The users' accuracy of urban land type is relatively high with 93.67% in 2010, 92.65% in 2015 and 91.32% in 2018 (Table 2). Overall, the urban land accuracy shows a decreasing trend, which resulted from the fuzzy and unidentifiable urban-rural boundaries owing to the continuous pattern of urban-rural land driven by China's fast urban development since the 21[st] century. In CLUD, the change polygons were identified based on the human interpretation. The validation of UIS and UGS fractions in each period showed that the RMSEs were 0.11–0.12 and 0.11–0.12 respectively, and the R values were 0.89–0.91 and 0.87–0.90, respectively (Table 3). The R and RMSE for the changed UIS areas during 2000-2018 are 0.88 and 0.12, respectively; and for the changed UGS areas during 2000-2018 are 0.85 and 0.12, respectively.

[Insert Table 2 here]

[Insert Table 3 here]

7.2 Patterns and dynamics of UIS and UGS since the beginning of the 21[th] century

Our result indicated that China's UIS increased from $2.46 \times 10^4$ km$^2$ in 2000 to $5.35 \times 10^4$ km$^2$ in 2018 (Fig. 7). From the perspective of the quality of dwellers' habitat environments, the percentage of UIS in China's urban area in 2018 is 74.42%, showing a higher UIS density than developed countries like the USA (Kuang et al., 2014). However, the UIS percentage in urban area decreased from 74.42% in 2000 to 68.21% in 2018 owing to the improvement of urban greening condition. As shown in Fig. 7, the UIS across China is mainly distributed in the coastal and central regions and relatively discrete in the western regions. The pattern of "high in east and low in west" of national UIS remained unchanged between 2000 and 2018 (Fig. 7). China's UGS shows an increasing trend. The total UGS area increased from $1.00 \times 10^4$ km$^2$ in 2000 to $1.83 \times 10^4$ km$^2$ in 2018 (Fig. 8). Looking at both UIS and UGS in urban areas, our results indicate a slight increase in UGS density and decrease in UIS density, which was resulted from strengthening urban greening since the 21[st] century. The UGS percentage rose from 18.91% in 2000 to 22.30% in 2018. As shown in Fig. 9, UIS and UGS of cities from coastal, northeastern, and southwestern China have high spatial heterogeneity in and showed the different urban expansion rate in past 28 years.

The large discrepancies of UIS and UGS percentage in urban area were exhibited among eastern, central, western and coastal zones. The coastal zone showed a remarkable increasing trend from 16.50% in 2000 to 21.66% in 2018 (Fig. 9 and Fig. 11). We also found that the urban greening condition was positively improved in Beijing at the same period, which resulted in the increase of UGS percentage and decrease of UIS percentage in urban area (Fig. 9). It means that urban habitat environment in coastal zone has become more liveable and comfortable, which is associated with the greening of parks and forests. We also found that the western cities have relatively low UGS percentage in urban areas, which has a 0.86% lower than average of China owing to the low greening condition (Fig. 9 and Fig. 11).

[Insert Figure 11 here]

7.3 Comparisons of the CLUD-Urban product with other datasets

We compared the vector boundaries of urban areas with the existing land-use products and found their obvious discrepancies because of the differences in data production, data source, resolution and definition of urban land-use types. The spatial resolutions of land-cover products range from 30 m to 1000 m. Fig.12 provides a comparison of a list of urban land datasets (see Table 4 for these datasets), showing that our product has better performance in delineating the detailed spatial patterns of intra-urban land cover, i.e. the composite of UIS and UGS (note: both the GHS Built and GlobaLand 30 products have only two years). The accuracy of urban boundaries from CLUD-Urban is over 92% and is basically inconsistent with that of impervious surface map (Zhang et al., 2020). Our dataset has a higher classification accuracy in urban boundaries than that of GHSL with 90.3%, FROM-GLC with 89.6%, HBASE with 88.0%, GlobeLand 30 with 88.4% and NUACI with 85.6%. Furthermore, our CLUD-Urban product can accurately delineate the spatial heterogeneity of UIS and UGS composites, which showed the R with 0.90 and 0.89, and RMSE with 0.11 and 0.11, respectively. In those existing datasets, the UIS and UGS composites can't be effectively decomposed at pixel scale (Fig. 12).

[Insert Table 4 here]

[Insert Figure 12 here]

## 8. Discussions

8.1 The mapping advantages integrated with human-computer interpretation and GEE platform

In mapping urban land use/cover change at national scale, two pivotal steps were required to segment the urban land, rural settlements, and industrial and traffic lands in periphery of cities for accurately acquiring the urban boundaries and to retrieve the UIS and UGS fractions at pixel scale. The urban boundaries are generally mapped using classification methods such as unsupervised classifiers, supervised classifiers, human-computer interpretation and other advanced algorithms (i.e. ANN, SVM and random forest) (Wu & Murray, 2003; Zhang et al., 2020). Among these methods, human-computer interpretation is widely regarded as a most accurate method in classifying urban land use/cover changes, especially in both detecting changing information and extracting vector polygons as whole geo-features. However, this method takes more time and manual labour

to digitalize a large number of polygons. The CLUD has an advantage for providing the accurate urban boundaries and is updated at an interval of every five or three years from 2000 to 2018.

Cities or towns were classified as a homogeneous unit in CLUD. We developed the UIS and UGS fractions to fill the data gap for the requirement of urban environmental management. Here we adopted the advantage of high accuracy and long-time series in mapping urban land from CLUD. Meanwhile we also utilized the highly efficient computation and large storage capacities on GEE platform. In mapping CLUD-Urban product, we proposed to quantitively retrieve the UIS and UGS fractions using random forest. Because we used advantages of manual interpretation and intelligent computation, the CLUD-Urban exhibits high accuracy and reliability in delineating urban land surface property.

8.2 The potential implications in promoting habitat environment and urban sustainability

CLUD-Urban product may effectively delineate the "built-up environment" of Chinese cities, especially the maps on surface imperviousness and greening condition (Kuang, 2020b). The CLUD-Urban can be applied to such fields as enhancing the quality of urban habitat environment, reducing urban heat island, and improving prevention of rainstorm and flood disaster (Huang et al., 2018). Our pervious study indicated that the thermal dissipation strength of forest canopy or lawns in cities may be assessed at the pixel scale and that the greening projects are more effective in alleviating urban heat island intensity (Kuang et al., 2015). The CLUD-Urban product also helps identify urban flood regulation priority areas based on ecosystem services approaches (Li et al., 2020).

The analysis of CLUD-Urban indicates unprecedented rate and magnitude of urban expansion since the 21[st] century. The low UGS of cities in western zones indicates the needs to promote the greening level (Kuang & Dou, 2020). The CLUD-Urban product can also be used to assess SDG targets such as the ratio of land consumption to population growth, average share of the built-up area that is open space for public use. Therefore, the CLUD-Urban can provide many potential applications in development of sustainable, liveable, and resilient cities.

8.3 Limitations of the method and dataset

Although state-of-the-art technologies and methodologies were applied to the development of CLUD-Urban (Dong et al., 2018; Kuang et al., 2020), improvement of mapping CLUD-Urban quality still exists. For example, the retrieval of UIS and

UGS was conducted as a prerequisite of CLUD, which focused on the pixel decomposition of UIS and UGS in urban areas. If the UIS and UGS fractions are parameterized to input into hydrological process model or urban climate, the settlement or impervious surface located in the outskirts of a city or rural areas are removed from CLUD. To address this issue, the first-level classification or second-level classification on CLUD should be utilized to merge with UIS and UGS using the method in our pervious publication (Kuang et al., 2020a). Mapping CLUD requires a large amount of labour and time that many interpreters are involved in this work. The extraction of urban boundaries might be subjective and there's a time lag in mapping UIS and UGS. It is needed to develop some advance tools to extract urban boundaries using automatic algorithms.

Fine urban land use/cover change mapping at national scale with high-resolution multi-source data may be developed in the aid of big-data and cloud platform (Gong et al., 2020). The development of a series of new algorithms and models are pivotal for improving the accuracy of datasets in retrieving urban boundaries and land-cover composites. However, the geo-knowledge is still essential for retrieving the high-quality dataset (Kuang et al., 2018). The CLUD-Urban can contribute to the development of sustainable cities, such as GEO and UN-Habitat in future.

**9 Data availability**

All data presented in this paper are available in https://doi.org/10.5281/zenodo.4034161 (Kuang et al., 2020a). This new version datasets include the UIS and UGS fractions with a 30-m spatial resolution in 2000, 2005, 2010 2015 and 2018. A detailed metadata description is provided, including contact information.

**10 Conclusion**

The CLUD-Urban – China's UIS and UGS fraction datasets with 30-m spatial resolution were generated using multiple data sources. CLUD-Urban provides detailed delineation of UIS and UGS components in China for the years of 2000, 2005, 2010, 2015 and 2018. Comparing to other products, the novelty of this dataset is to take cities as heterogeneous units at the pixel level, which is consisted of UIS, UGS, and others. The accuracy of the CLUD-Urban dataset is over 92.65% using the integrated approach of visual interpretation and prior knowledge. The RMSEs of UIS and UGS fractions are 0.10 and 0.14, respectively. Results from the analysis of urban areas, including UIS and UGS, show large regional differences in China. CLUD-Urban provides fundamental data sources for examining urban environment issues and for delineating intra-urban structure or urban landscape at the national scale.

## Author contribution

KW, ZS and LX designed the research; ZS and LX implemented the research; KW, ZS and LD wrote the paper.

## Competing interests

The authors declare no conflict of interest.

## Acknowledgments

This study was supported by National Natural Science Foundation of China (NSFC) (41871343) and Strategic Priority Research Program A of the Chinese Academy of Sciences (XDA23100201). We thank Master Yali Hou and Changqing Guo for processing the data, and Dr. Fengyun Sun and Dr. Rafiq Hamdi for their help in manuscript editing.

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

**Table 1**: **The multitemporal data series used in this research**

| Year | Path | Row | Sensor | Spatial resolution (m) |
|---|---|---|---|---|
| 2000 | | | Landsat TM | 30 |
| 2005 | | | Landsat TM | 30 |
| 2010 | 118–149 | 23–43 | Landsat TM/ETM+, HJ-1, CBERS -1 | 30 |
| 2015 | | | Landsat 8 OLI | 30 |
| 2018 | | | Landsat 8 OLI | 30 |

| Data sources | Resolution |
|---|---|
| SRTM Digital Elevation Model data | 30 m |
| NDVI | 30 m |
| Google Earth images | 0.6 m |

 Table 2: Confusion matrix of the China Land-Use/Cover Dataset

| Year | Land type | Samples size | Accuracy for specific land type | | | Source |
|---|---|---|---|---|---|---|
| | | | Producers' accuracy (%) | Users' accuracy (%) | Overall accuracy | |
| 2000 | Built-up area | 8,055 | | | 98.92% | (Zhang et al. 2014) |
| 2005 | Built-up area | 7,382 | | | 97.01% | (Zhang et al. 2014) |
| 2010 | Built-up area | 7,875 | - | - | | (Kuang, Liu, Dong, Chi, & Zhang 2016) |
| | Urban land | | 94.30 | 93.67 | | |
| | Rural settlement | | 91.76 | 91.76 | | |
| | Industrial and traffic lands | | 91.67 | 90.26 | | |
| 2015 | Built-up area | 7,235 | - | - | | This Study |
| | Urban land | | 91.30 | 92.65 | | |
| | Rural settlement | | 89.29 | 93.28 | | |
| | Industrial and traffic lands | | 95.45 | 91.30 | | |
| 2018 | Built-up area | 7,235 | - | - | | |
| | Urban land | | 90.40 | 91.32 | | |
| | Rural settlement | | 88.19 | 92.18 | | |
| | Industrial and traffic lands | | 94.43 | 92.13 | | |

**Table 3: Accuracy assessments for the UIS and UGS fractions.**

| Year | UIS | | UGS | |
|------|-----|------|-----|------|
| | R | RMSE | R | RMSE |
| 2000 | 0.91 | 0.11 | 0.90 | 0.11 |
| 2005 | 0.90 | 0.11 | 0.90 | 0.11 |
| 2010 | 0.90 | 0.11 | 0.88 | 0.11 |
| 2015 | 0.91 | 0.11 | 0.88 | 0.11 |
| 2018 | 0.89 | 0.12 | 0.87 | 0.12 |

**Table 4: A summary of existing urban land products.**

| Name | Spatial resolution | Abbreviation | Method | Reference |
|------|--------------------|--------------|--------|-----------|
| Land Cover from Moderate-resolution Imaging Spectroradiometer | 500m | MODIS LC | Decision tree classification | (Friedl et al., 2010) |
| European Space Agency global land-cover data | 300m | ESA LC | Unsupervised classification and change detection | (Bontemps et al., 2011) |
| Built-up grid of the Global Human Settlement Layer | 30m | GHS Built | Symbolic machine learning | (Pesaresi et al., 2013) |
| Global Land Cover at 30m resolution | 30m | GlobaLand30 | Pixel-Object Knowledge (POK)-based classification | (Chen et al., 2015) |
| Multi-temporal Global Impervious Surface | 30m | MGIS | Normalized urban areas composite index | (Liu et al., 2018) |
| Annual maps of global artificial impervious area | 30m | GAIA | "Exclusion/Inclusion" approach | (Gong et al., 2020) |


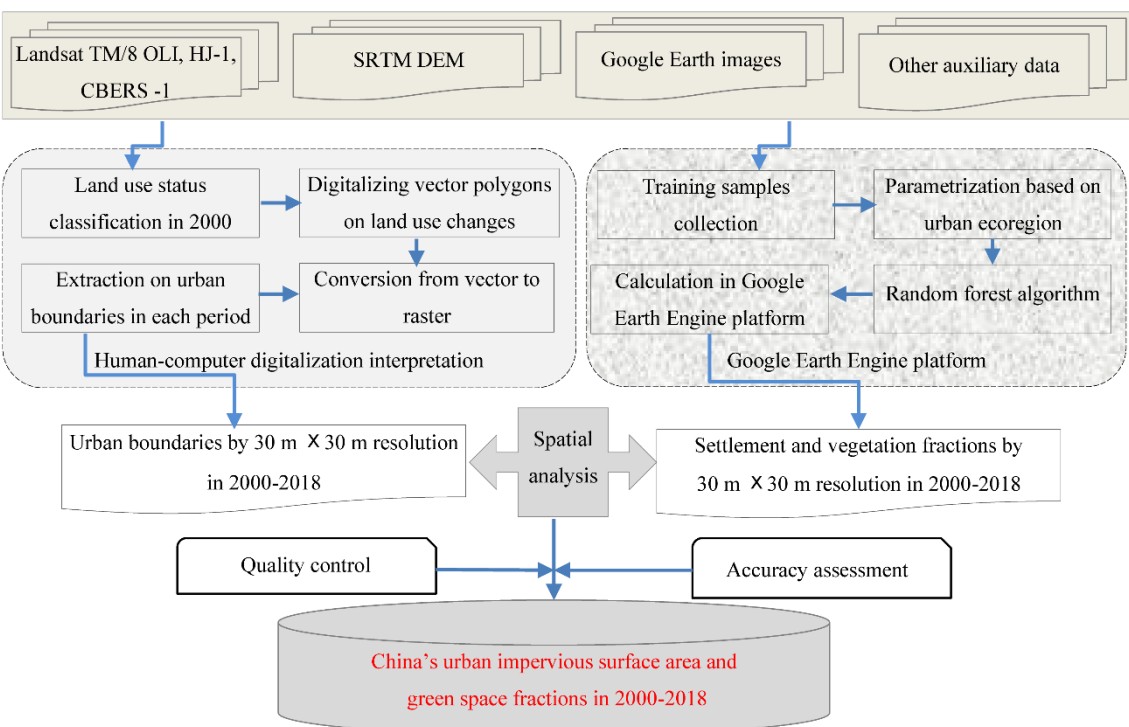

**Figure 1**: **The technological flowchart of generating CLUD-Urban product.**

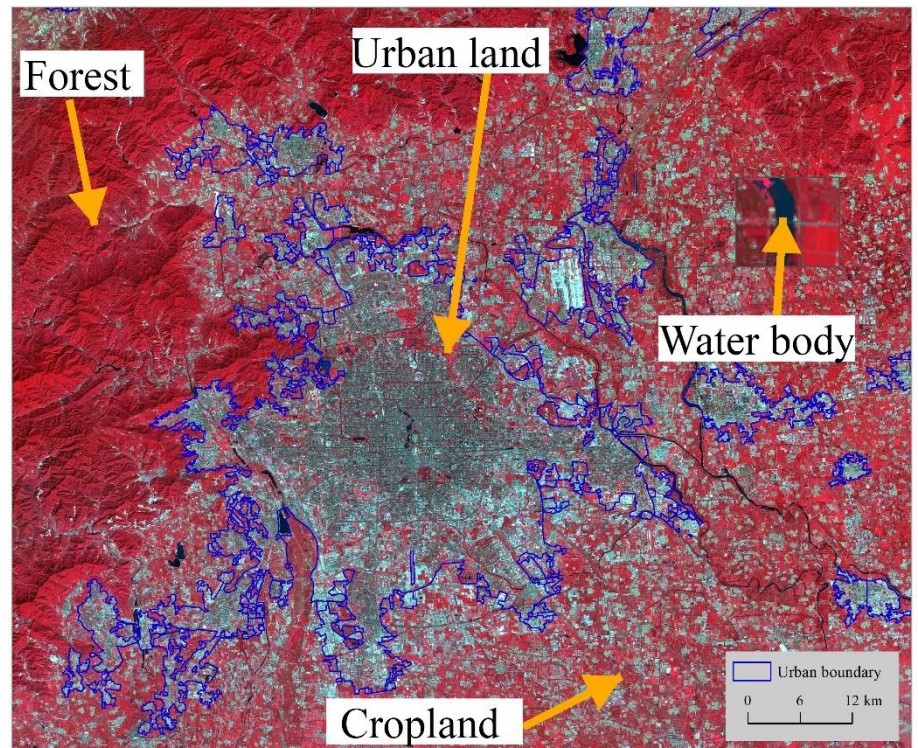

**Figure 2: The interpretation symbols and extracted urban boundaries from Landsat images in Beijing city. (The images were provided by Geospatial Data Cloud site, Computer Network Information Center, Chinese Academy of Sciences**
**(http://www.gscloud.cn).**

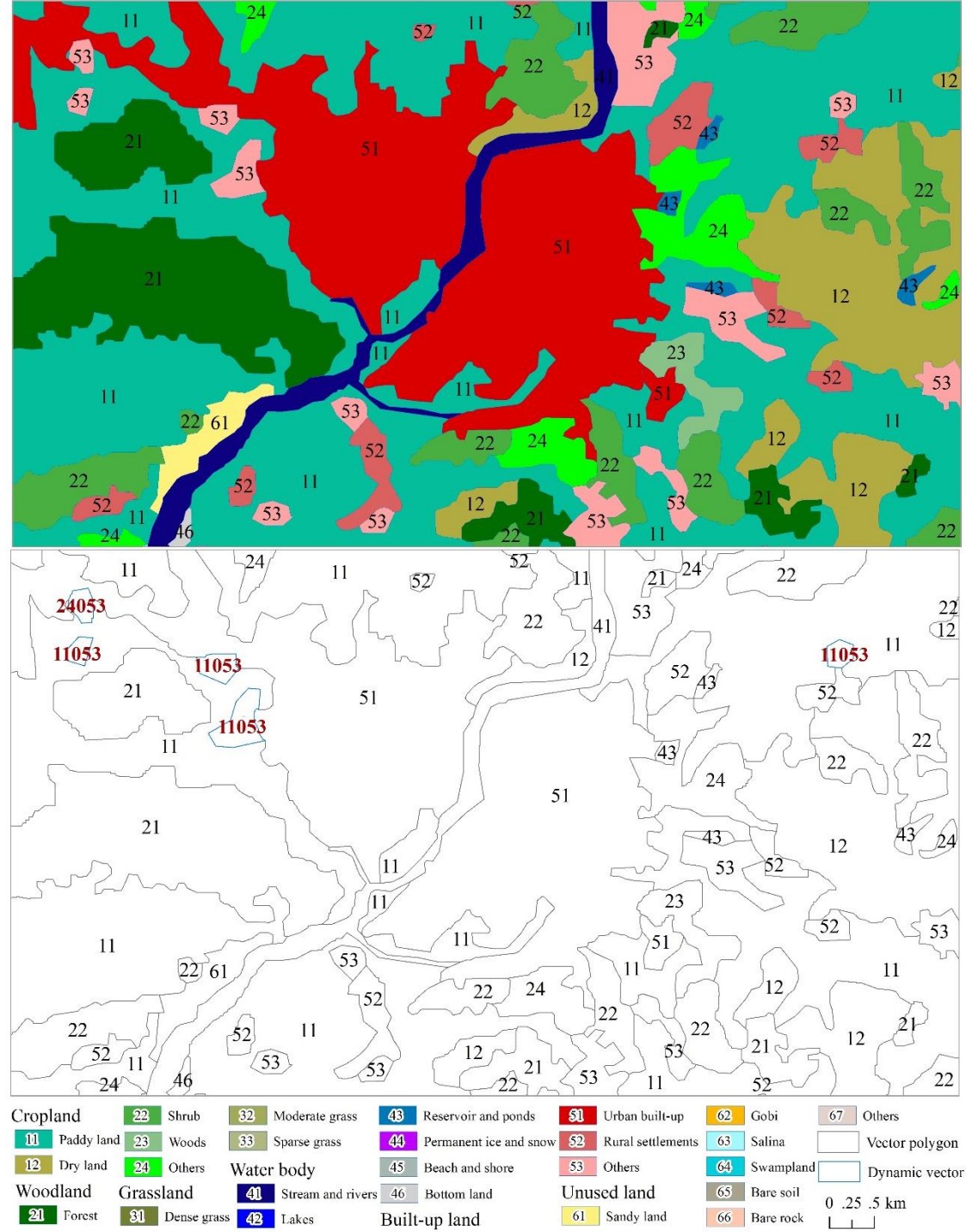

**Figure 3: Land use classification and extracted vector polygons as an example with Conghua district of Guangzhou city.**


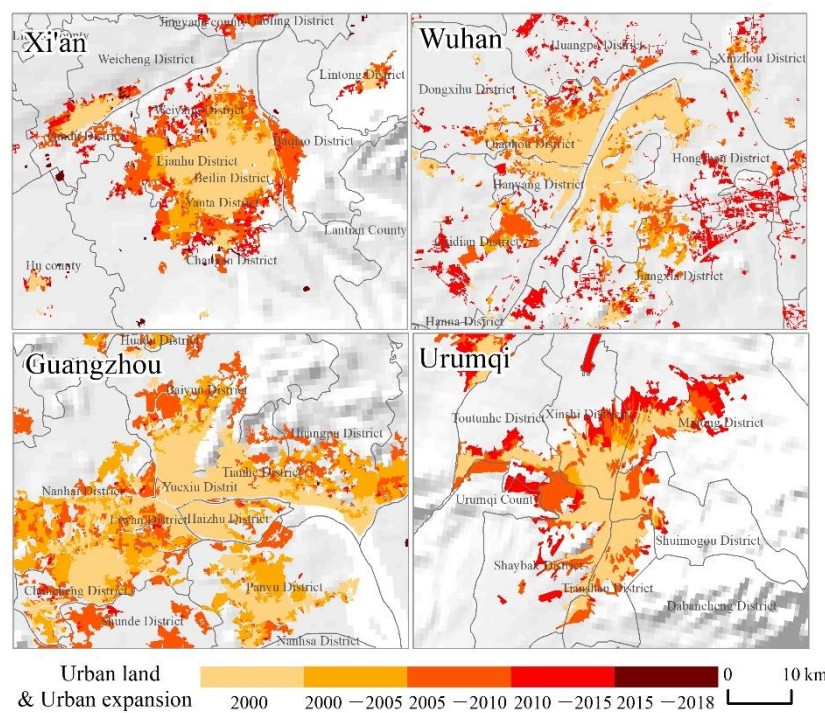

**Figure 4: The urban boundaries extracted from CLUD with 30-m resolution in selected cities. (The administrative boundaries were provided by National Geomatics Center of China (http://www.webmap.cn); DEM dataset was downloaded from SRTM 90 m Digital Elevation Data (http://srtm.csi.cgiar.org/))**

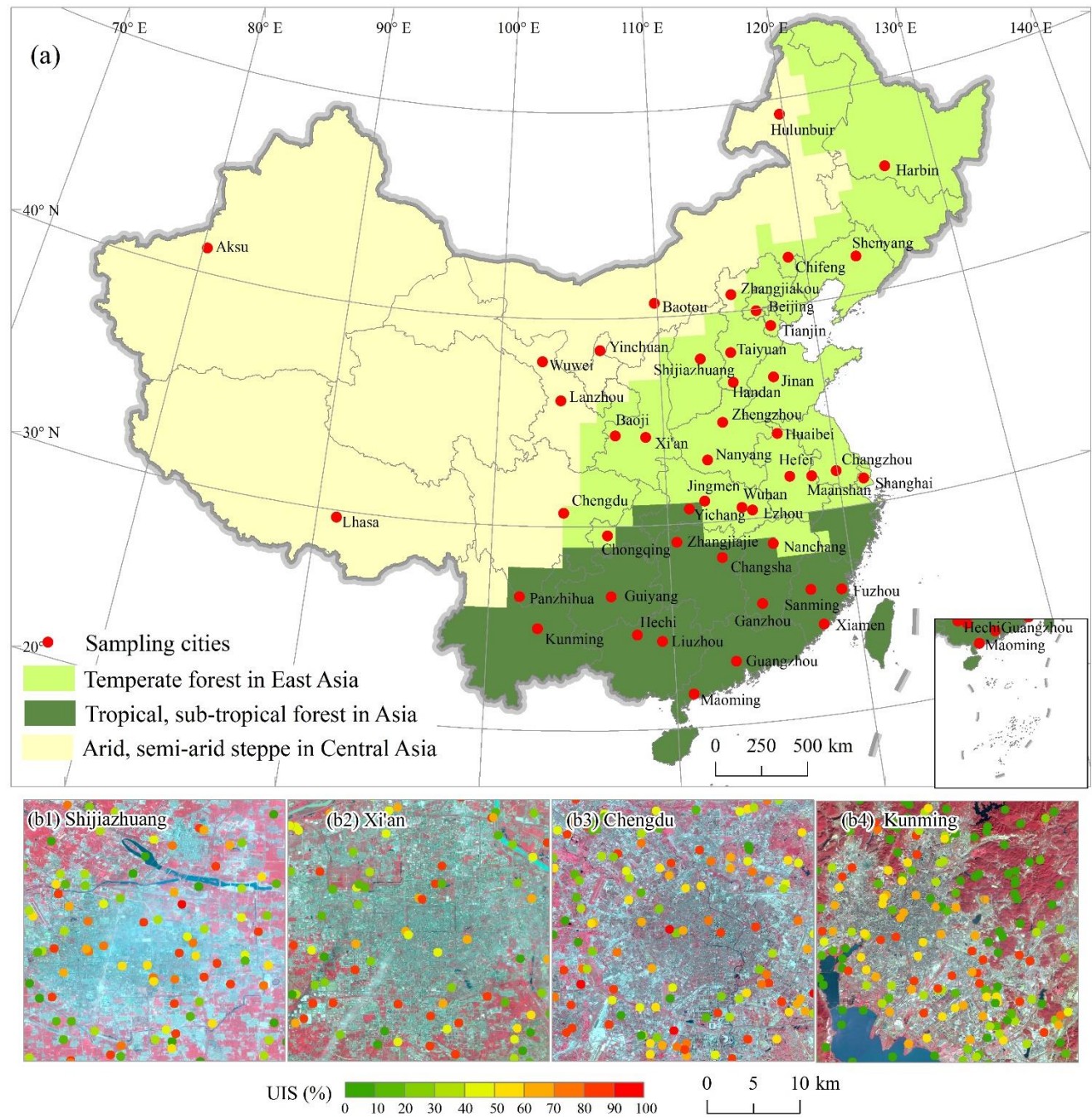

**Figure 5: Distribution of sampling cities in China and training samples in selected cities. (The images were provided by Geospatial Data Cloud site, Computer Network Information Center, Chinese Academy of Sciences (http://www.gscloud.cn). The administrative boundaries were provided by National Geomatics Center of China (http://www.webmap.cn))**

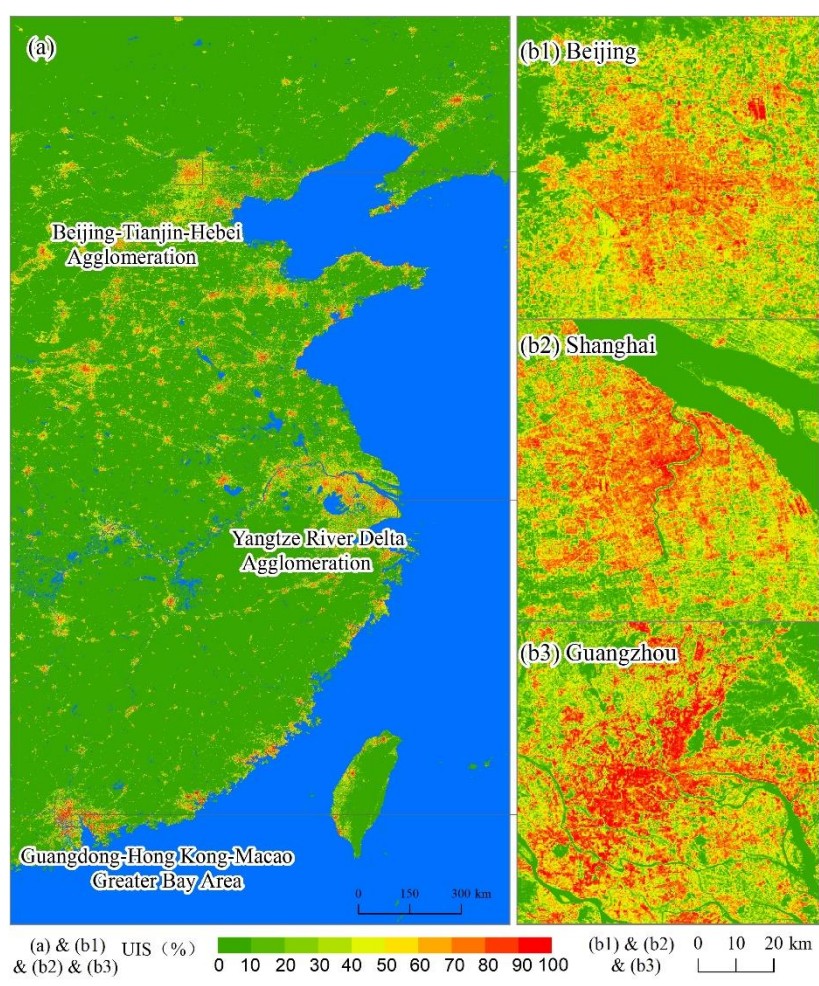


**Figure 6**: **Distribution of sampling cities in China and training samples in selected cities. (The administrative boundaries and residential points information were provided by National Geomatics Center of China (http://www.webmap.cn))**


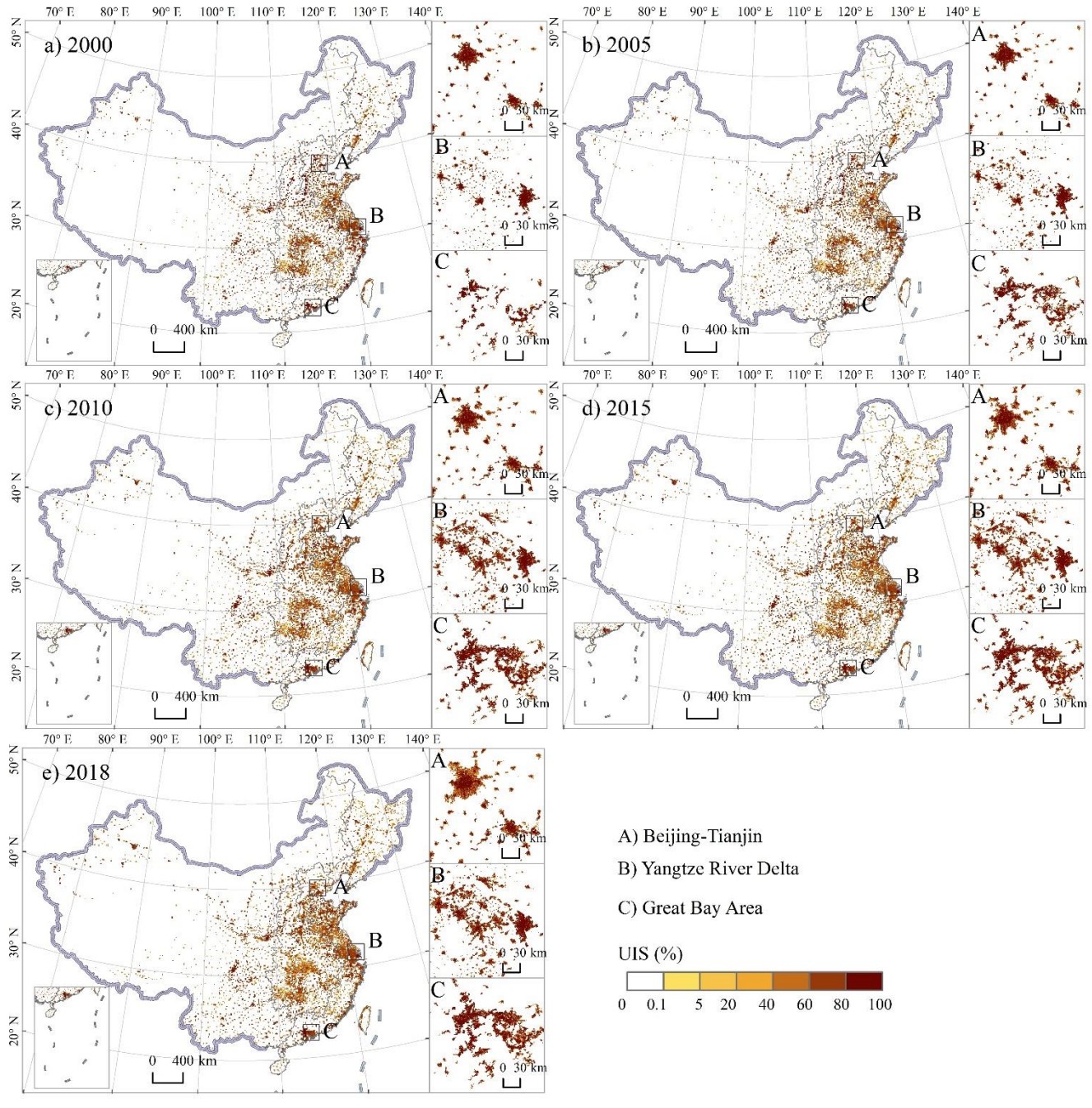

**Figure 7**: **Spatial distribution of urban impervious surface (UIS) in 2000–2018 across China. (The administrative boundaries were provided by National Geomatics Center of China (http://www.webmap.cn))**

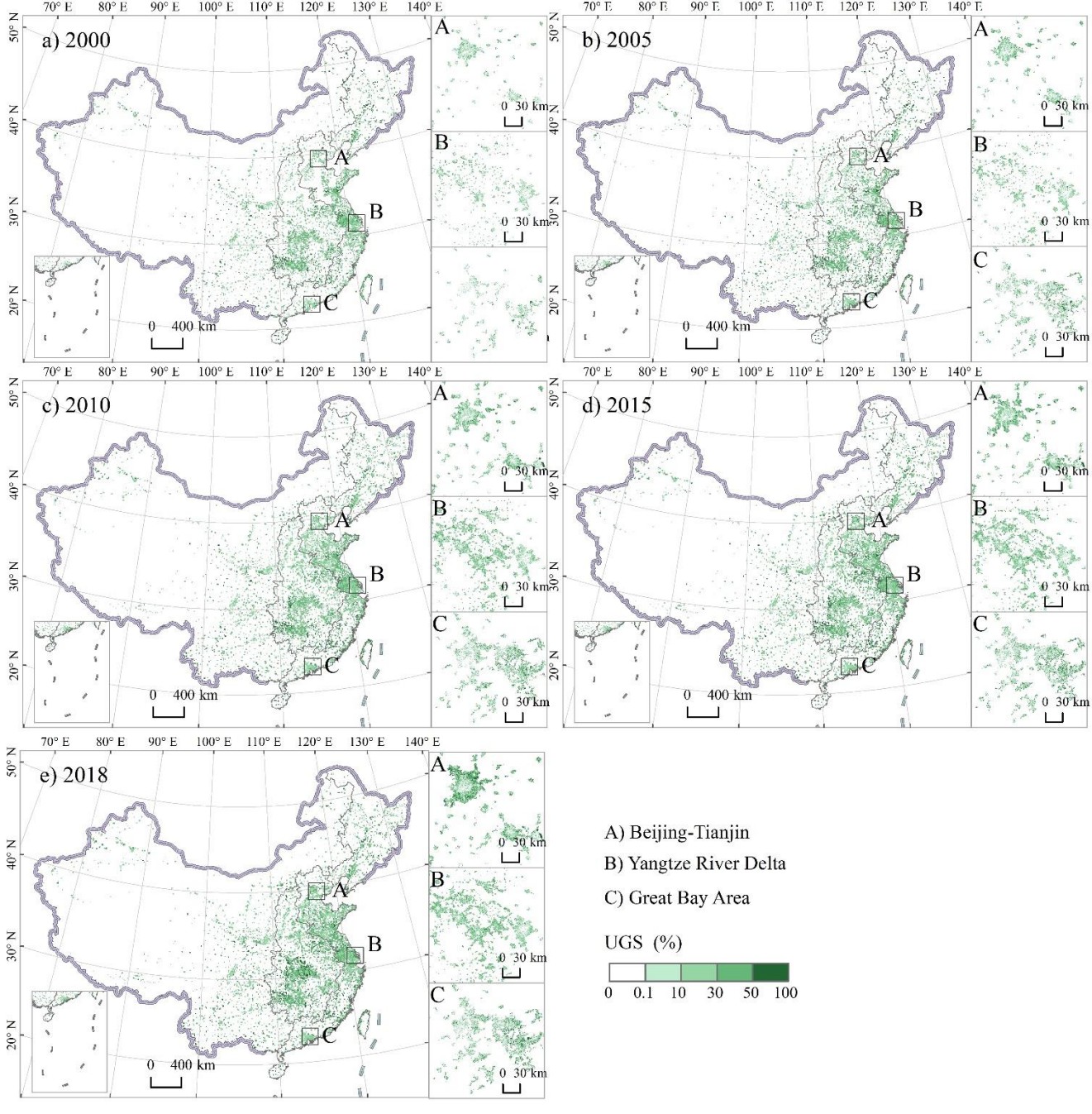

**Figure 8**: **Spatial distribution of urban green space (UGS) in 2000–2018 across China. (The administrative boundaries were provided by National Geomatics Center of China (http://www.webmap.cn))**

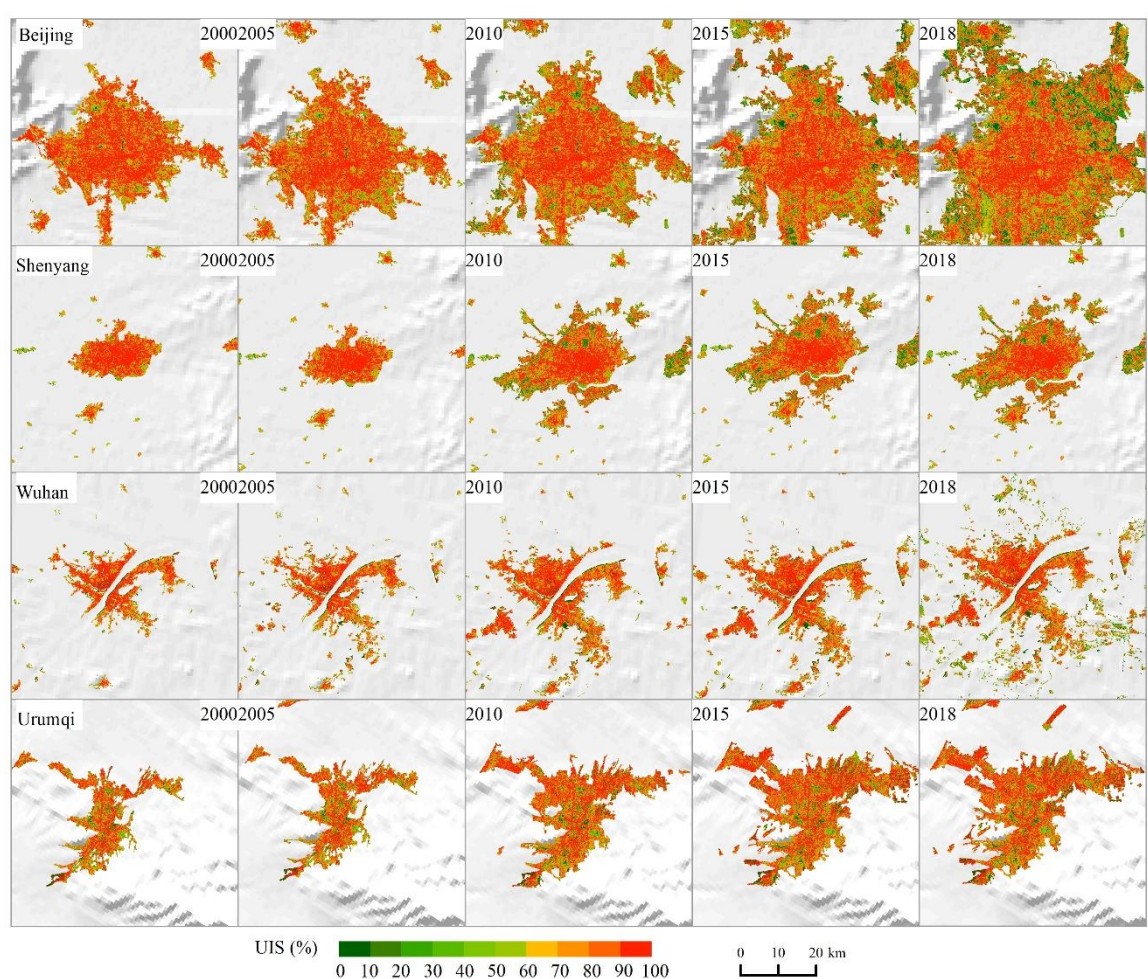



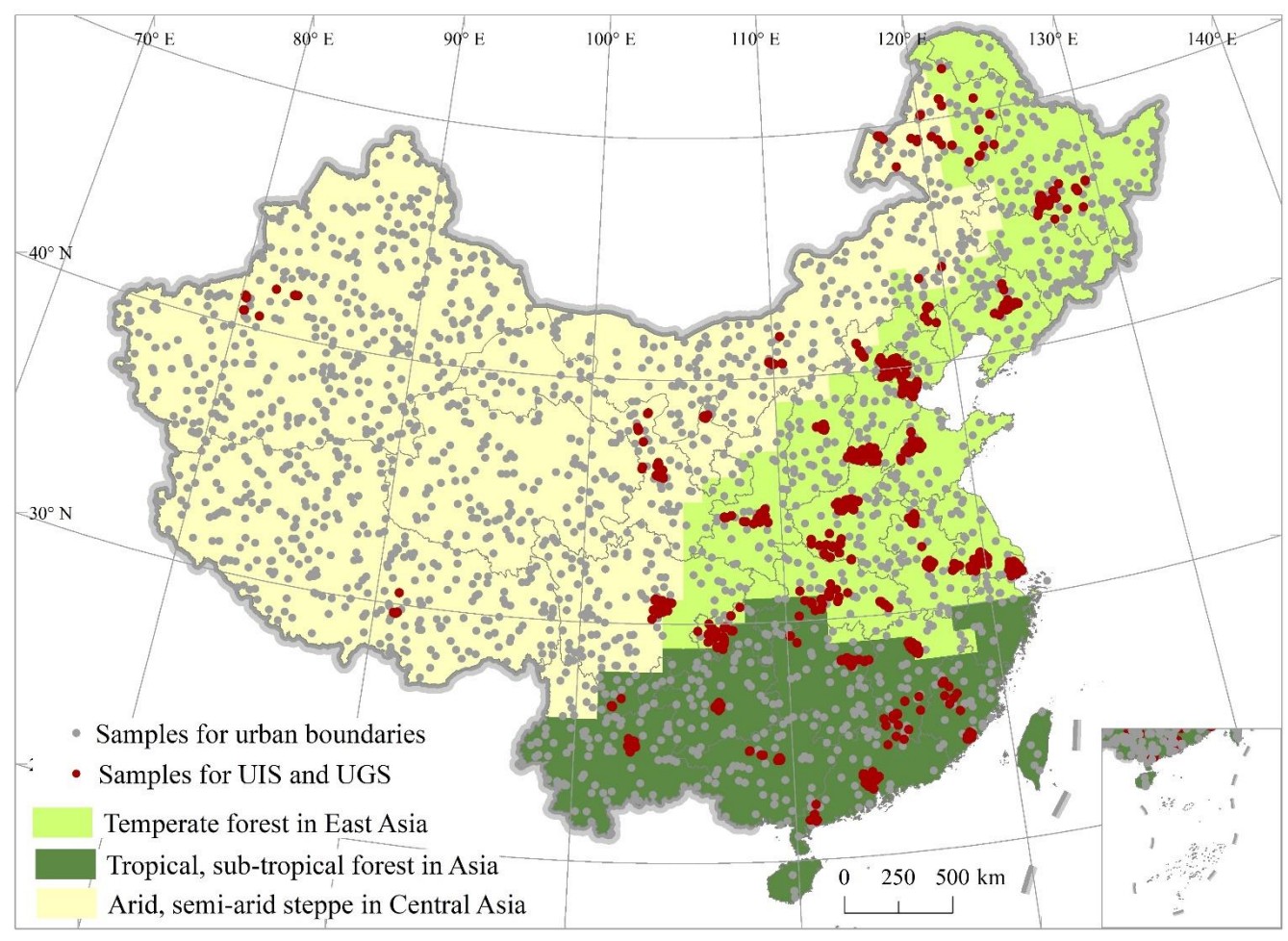

**Figure 10**: **Validation samples on CLUD-Urban product. (The administrative boundaries were provided by National Geomatics Center of China (http://www.webmap.cn))**

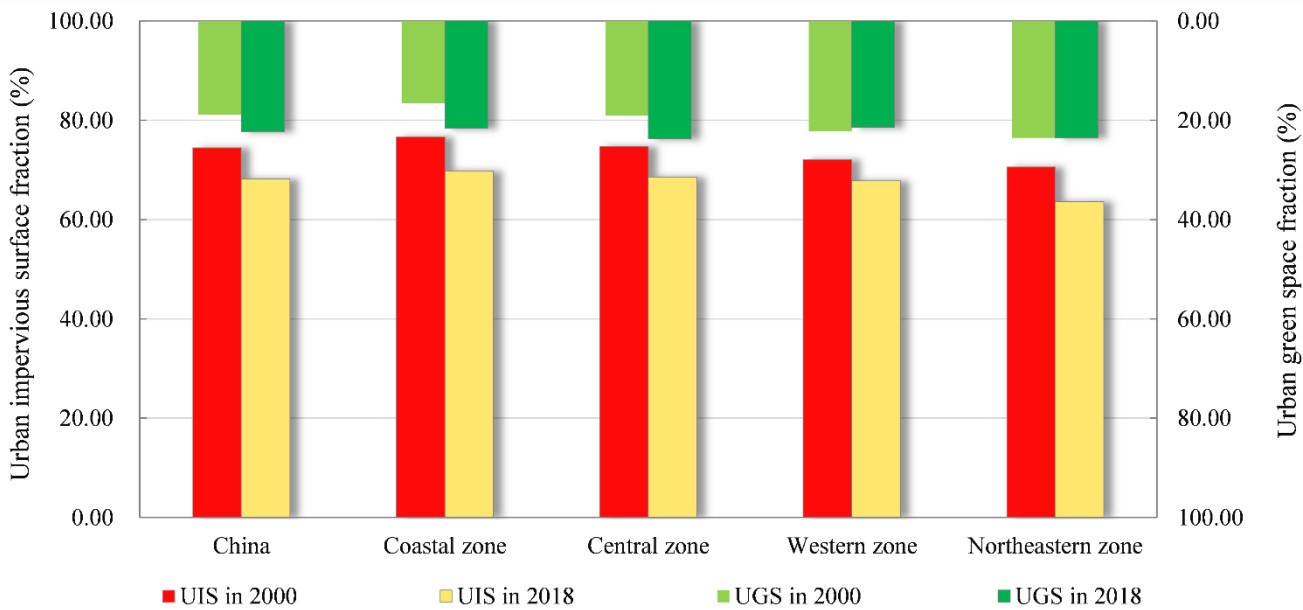

**Figure 11: The urban impervious surface (UIS) and urban green space (UGS) fractions at national and regional scales (coastal, central, western and northeastern zones) in 2000 and 2018.**


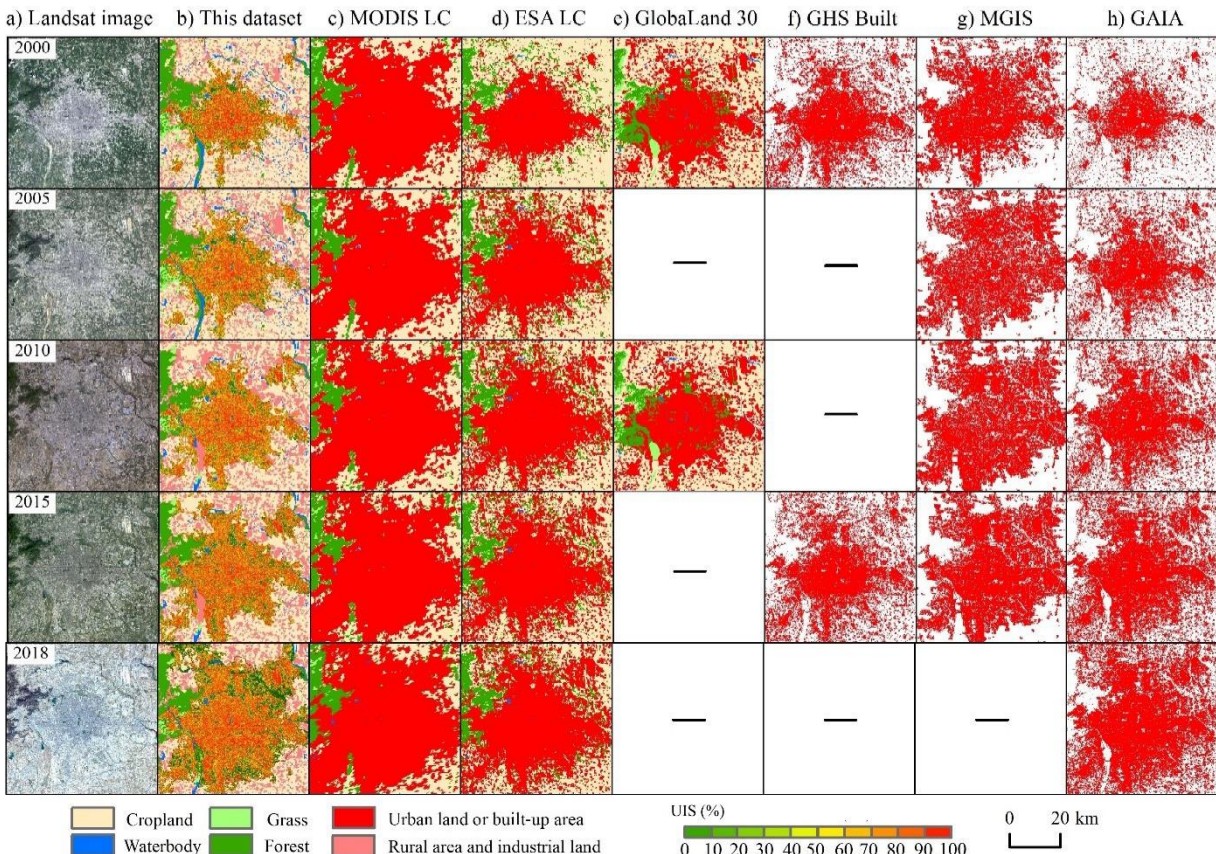

**Figure 12: A comparison of urban land cover between this product and other datasets in Beijing (The Landsat images were provided by Geospatial Data Cloud site, Computer Network Information Center, Chinese Academy of Sciences** (http://www.gscloud.cn))