# Peer review of "A 30-meter resolution dataset of China's urban impervious surface area and green space fractions, 2000–2018"

_Earth System Science Data, 2020_

## Referee Comment (RC1) · Anonymous Referee #1 · 20 Jul 2020

In this study, a multi-source data-based method for mapping UISA and UGS fractions in China using Google earth engine was proposed, and datasets for 2000-2018 were obtained. In the subpixel scale, a pixel of 30m*30m is regarded as a combination of UISA, UGS and others. The topic of the study is interesting and fits the scope of the journal. In this dataset, the composition of urban landscape is described at a more detailed scale, which makes up for the lack of data in China. However, there are still some problems that need more explanation. What's more, the innovation of this study is not clearly stated, which should be highly improved.

General comments: 1. What is the main innovation of this research? This directly

determines the value of this research. Compared with existing datasets of the same type, such as the NLCD dataset mentioned in this paper, what are the differences and improvements in the calculation method? Or does it just fill in the gap of this data in China? 2. Have you considered the unification of images of different years and the unification of images of different satellites, such as China China-Brazil Earth Resources Satellite (CBERS-1) and Huan Jing (HJ-1A/B) satellite with Landsat? I suggest more introduction of data processing. 3. When calculating the UGS fraction, have you considered the different vegetation types? Like the difference between trees and grass? Will this make a difference to the results?

Specific comments: Line 27 on page 1: It should be "environment" since it refers to the overall state of environment. Line 31 on page 1: Does rapid urbanization process result in rapid increase in urban green space? Are there any references supporting this claim? Line 34 on page 2: "other" should be deleted since China is a developing country, not a developed country. Line 43 on page 2: Does this sentence mean the definitions of different products for urban areas are based on IGBP or FAO? Line 49 on page 2: I think it's more likely to be cause and effect. So, it should not be "furthermore" here. Line 61 on page 2: The expressions of urban landscape and urban landscape have appeared for many times. The usage of this phrase is different. Please unify the form of this expression. Line 82 on page 3: When CLUD first appears in the text, a full name is required. Line 94 on page 3: "as well as" should not be used here because cultural services are part of the ecosystem services. Line 95 on page 3: Is the "restoration" here a kind of cultural services? How to understand? Line 96 on page 3: What does the "exclude this component" mean? Most products do not distinguish between parks, trees and grass? Line 97 on page 4: "a" should be deleted. Line 101-102 on page 4: In extremely dense urban agglomerations like the Yangtze River Delta, the boundaries between some cities are not obvious. How to deal with this? Are there any problems? Line 106 on page 4: The urban impervious surface area and the urban impervious surface area fraction are both abbreviated UISA. So as the UGS. This statement is ambiguous. Please modify it. Maybe you can use UISAF and

UGSF to present the fractions. Line 116 on page 4: Is it possible to use probabilities to represent ratios? How do you justify this logic? Are the input UISA classification values from pure pixels or mixing pixels? Line 124 on page 4: It should not be ith here. i should be a total number, or it should be expressed as n. Line 146 on page 5: How many samples were surveyed in the field? Line 151 on page 6: What is the higher resolution? Does visual interpretation take the smallest unit of Google images as a single pixel to calculate the number of impervious and vegetation units? Line 153 on page 6: "a" should be deleted. Line 154 on page 6: "densities" would be better to be "fractions". Line 155 on page 6: It should be "values in the same area were". Line 157 on page 6: "shows" should be "showed". Line 157 on page 6: There are two ".". Line 158 on page 6: It should be ", respectively" and "validation of". Line 167 on page 6: It should be a new sentence form "note". Line 169-171 on page 6: How is the urban area defined in this study? Are you using existing data and method? If so, it can not prove the advantages of this study. Line 174 on page 6: What is the actual urban expansion rate? Can you give a value to prove the similarity? Line 175 on page 6: "other" here should also be deleted. Line 181 on page 6: Does the UGS here refer to urban green space or areas with high green space fraction? Line 184 on page 7: The "main urban areas" here may not be a very appropriate statement. Line 188 on page 7: Since there are other components, why don't you say high proportional UGS represents parks and greenbelts with ecological functions? Line 197 on page 7: "was" should be "were". Line 199 on page 7: Please be consistent with the previous. Determine to use "dataset" or "datasets" to express the UISA and UGS data? Table 1: "Note" should be left aligned. Table 4: There should be a "Note" before "MRE...". Figures: All maps lack a compass. Figure 4: What are the meanings of the small pictures on the right? Please make more explanations. Figure 8: There are two sets of legends in the figure, and some colors are similar. How to distinguish them?

---

## Referee Comment (RC2) · Anonymous Referee #2 · 10 Aug 2020

The manuscript provides the China's 30-m UISA and UGS fraction datasets based on the urban area in CLUDs by logistic regression and linear calibration using NDVI from Landsat data.

I would suggest the authors reorganize the sections, give more details on the samples and mapping algorithm, discuss more about the accuracy of the maps and comparison with various datasets, add an discussion part and resubmit this paper.

Figures:

1. For Figure 5 and 8, it would be better to remove the other land cover types and only show the fraction of UISA. The color is confusing among the vegetation types and the lower percentage of UISA.

Introduction:

2. There are quite a few existing dataset/report that are providing information about urban green spaces and urban land use categories of China. For examples, (1) https://www.mdpi.com/2072-4292/10/10/1569/htm, (2) https://www.sciencedirect.com/science/article/pii/S2095927319307054?via%3Dihub

Method

3. I feel the methodology section can be written to make it clearer (e.g., sample selection, the retrieval of UGS fraction)
4. Effect of urban boundary. How to define the urban area and extract the urban boundary are not clear? There are some other datasets providing the urban extent using different algorithms and data sources, e.g.:

   *Gong P, Li X C, Zhang W. 40-Year(1978-2017) human settlement changes in China reflected by impervious surfaces from satellite remote sensing. Science Bulletin, 2019, 64,https://doi.org/10.1016/j.scib.2019.04.024*

   *Zhou, Y., Li, X., Asrar, G. R., Smith, S. J., & Imhoff, M. (2018). A global record of annual urban dynamics (1992–2013) from nighttime lights. Remote Sensing of Environment, 219, 206-220.*

   *Li, X., Gong, P., Zhou, Y. et al. 2020. Mapping global urban boundaries from the global artificial impervious area (GAIA) data. Environmental Research Letters. https://iopscience.iop.org/article/10.1088/1748-9326/ab9be3/meta*

5. In model training, 28 capital cities were selected to extract samples for LRM model input. Are these capital cities capable to represent other cities in China? As is mentioned, the UISA is related with economic and geographic conditions, but the capital cities are commonly the better developed region than the other cities.
6. Does the urban land changes area refer to the area with land conversion between urban area and other land cover types (cropland to urban) or the changes within urban area (from built-up to greenspace)?
7. Why not use samples in 90m×90m for validation?
8. Would you please provide the details of validation samples, e.g., spatial distribution, types.

Results

9. Currently, there is no discussion part. What is the potential application and the uncertainty of this datasets? And the results are too short and simplified. Please add more details such as comparisons with other UGS, UISA dataset, line graphs of the temporal changes of UISA in different regions to support the conclusion of "high in east and low in west".

---

## Referee Comment (RC3) · Anonymous Referee #3 · 10 Aug 2020

This paper developed a 30-m resolution dataset of China's urban impervious surface area and green space fraction from 2000 to 2018. The fraction information of impervious and green space is revealed from remotely sensed indicators. I have some comments on the adopted approaches and presented results. Main comments (1) It is difficult to derive the logistic regression model directly since most of the available observations are limited in this study (i.e., only five observations). Why not use continuous observations (i.e., annual) to fit the logistic regression model. (2) Retrieving the ISA information from the NDVI directly seems not reasonable. First, the maximum NDVI used in this study may be fluctuated over years and across spaces. Second, for the built-up areas in arid regions, the proposed approach of estimating the impervious

surface information from the NDVI is not reliable. This is an issue that needs to be well addressed. (3) The comparison of urban areas with other products is necessary, including the accuracy and urban area. It is inadequate if only presenting these comparable figures here directly. (4) A similar approach to estimate the fraction of green space is not reasonable also. I am wondering if this linear relationship can well estimate the green space. (5)

Minor comments: Page 3, Line 78-80: how to harmonize the spectral bands between Landsat and HJ (or CBERS-1)? Page 3, Line 95: the author mentioned that most classifications exclude the green space, which is not accurate. Green space is a kind of definition from land use, which is consists of trees, shrubs, grasses etc., which are general land cover types. Page 4, Line 100: the definition of new and old urban lands from their colors still needs more evidence. Visually, the new built-up areas such as residential which has similar layout and materials may be similar to the old urban lands Page 8, Line 230: there is a repeated reference (Dong et al., 2017).

---

## Referee Comment (RC4) · Anonymous Referee #4 · 10 Aug 2020

The authors present multi-year maps of urban imperviousness and greenness of China, which were estimated based on hand-drawn urban boundary and the relationship between vegetation greenness and surface imperviousness. Despite the data might be valuable to a variety of urban-related applications, there are many uncertainties remain. As a data set, these uncertainties should be clearly addressed so that users could better use it. First, using NDVI as the only indicator to estimate surface imperviousness is problematic. The NDVI-based method would overestimate the extent of impervious surfaces because of their similar characteristics as some land uses/covers on NDVI images, especially bare ground. This is especially true in most Chinese cities as they have seen substantial expansions during the study period and

the extent of bare ground cannot be ignored. Second, calibration of NDVI-ISA relationship is not clear in many aspects. For example, how was ISA reference measured for model calibration? What was the performance of region averaged model compared to city-specific ones? Was the model calibrated once and applied through time or annually? Third, the modeling was based on an existing product (i.e., CLUD), which was based on visual interpretation if I am correct). More details about how urban boundary was extracted and updated should be stated. Without this information, it is hard for readers to know whether urban expansion captured by CLUD was true urbanization or just hand-drawn inconsistency. How was the accuracy of CLUD assessed? Because the definition of urban in CLUD is more based on administrative perspective instead of surface imperviousness, I want to know more how accuracy of 92-99% was calculated (Lines 149-150). Last, data uncertainties and limitations should be further addressed. For example, what are spatial and temporal accuracy variations? How consistent was the estimation over time (i.e., is it reliable to use this data set to capture real ISA change)?

————————————————————

---

## Author Comment (AC1) · 17 Sep 2020

Thank you for the comments and suggestions. These comments were very helpful for revising and improving our paper. We have responded to the comments point by point and made the detailed revisions embedded in the manuscript with the line numbers indicated in the responses.

**Comment 1:** In this study, a multi-source data-based method for mapping UISA and UGS fractions in China using Google earth engine was proposed, and datasets for 2000-2018 were obtained. In the subpixel scale, a pixel of 30m ✕ 30m is regarded as a combination of UISA, UGS and others. The topic of the study is interesting and fits the scope of the journal. In this dataset, the composition of urban landscape is described at a more detailed scale, which makes up for the lack of data in China. However, there are still some problems that need more explanation. What's more, the innovation of this study is not clearly stated, which should be highly improved.

**Response:** Thank you for the constructive comments. Urban impervious surface (UIS) and urban green space (UGS) are two core components for characterizing urban underlying environments. However, the UIS and UGS often are mosaicked in the urban landscape with complex structures and composites. Therefore, the 'hard classification' or binary single type cannot be effectively used to delineate spatially explicit urban land surface property. Although the six mainstream datasets on global or national urban land use/cover products with 30-m resolution have been developed, they only provide the binary pattern or dynamic of a single urban land type, which cannot effectively delineate the quantitative components or structure of intra-urban land cover. Here we proposed a new mapping strategy to acquire the multitemporal

and fractional information of the essential urban land cover types at national scale through synergizing the advantage of both big data processing and human interpretation in aid of geo-knowledge.

   **Changes in manuscript:** In the revised version, we highlighted the innovation of this work in the Introduction section in L75-100. We also discussed the advantages of this method and CLUD-Urban product in discussion section in L285-305.

**Comment 2:** General comments: 1. What is the main innovation of this research? This directly determines the value of this research. Compared with existing datasets of the same type, such as the NLCD dataset mentioned in this paper, what are the differences and improvements in the calculation method? Or does it just fill in the gap of this data in China?

**Response:** Thank you for your comments. Cities or towns were classified as a homogeneous feature in original CLUD. In this research, we developed the UIS and UGS fractions to fill the data gap from the requirement of urban environmental management. Here we adopted the advantage of high accuracy and long-time series in mapping urban land from CLUD. Meanwhile we utilized the highly efficient computation and large storage capacities of GEE platform. In mapping CLUD-Urban product, we proposed to quantitively retrieve the UIS and UGS fractions using random forest. The assessment results indicated the higher accuracy of the CLUD-Urban than NLCD UIS. Our product has high reliability owing to using the advantage of manual interpretation and intelligent computation.

**Changes in manuscript:** We rewrote the method part and added some discussions on this issue in L105-225 and L285-330.

**Comment 3:**2. Have you considered the unification of images of different years and the unification of images of different satellites, such as China China-Brazil Earth Resources Satellite (CBERS-1) and Huan Jing (HJ-1A/B) satellite with Landsat? I suggest more introduction of data processing.

**Response:** China-Brazil Earth Resources Satellite (CBERS-1) and Huan Jing (HJ-1A/B) satellite images were only used in extracting the vector polygons of CLUD in 2010. We added more texts to describe the data processing in China China-Brazil Earth Resources Satellite (CBERS-1) and Huan Jing (HJ-1A/B) satellite images.

**Changes in manuscript:** We revised the data processing on satellite images in L120-125.

**Comment 4:** 3. When calculating the UGS fraction, have you considered the different vegetation types? Like the difference between trees and grass? Will this make a difference to the results?

**Response:** Yes, we considered the difference on trees and grass. In mapping green spaces fraction, the training samples on trees and grass in urban areas were selected to input into parameter in random forest model.

**Changes in manuscript:** We explained this issue in L170-180.

**Comment 5:** Specific comments: Line 27 on page 1: It should be "environment" since it refers to the overall state of environment. Line 31 on page 1: Does rapid urbanization process result in rapid increase in urban green space? Are there any references supporting this claim? Line 34 on page 2: "other" should be deleted since China is a developing country, not a developed country.

**Response:** Thank you for the specific suggestions. We revised those sentences.

**Comment 6:** Line 43 on page 2: Does this sentence mean the definitions of different products for urban areas are based on IGBP or FAO? Line 49 on page 2: I think it's more likely to be cause and effect. So, it should not be "furthermore" here. Line 61 on page 2: The expressions of urban landscape and urban landscape have appeared for many times. The usage of this phrase is different. Please unify the form of this expression. Line 82 on page 3: When CLUD first appears in the text, a full name is required. Line 94 on page 3: "as well as" should not be used here because cultural services are part of the ecosystem services. Line 95 on page 3: Is the "restoration" here a kind of cultural services? How to understand?

**Response:** Thank you for the specific suggestions. We revised those sentences.

**Comment 7:** Line 96 on page 3: What does the "exclude this component" mean? Most products do not distinguish between parks, trees and grass? Line 97 on page 4: "a" should be deleted. Line 101-102 on page 4: In extremely dense urban agglomerations

like the Yangtze River Delta, the boundaries between some cities are not obvious. How to deal with this? Are there any problems? Line 106 on page 4: The urban impervious surface area and the urban impervious surface area fraction are both abbreviated UISA. So as the UGS. This statement is ambiguous. Please modify it. Maybe you can use UISAF and UGSF to present the fractions. Line 116 on page 4: Is it possible to use probabilities to represent ratios? How do you justify this logic? Are the input UISA classification values from pure pixels or mixing pixels? Line 124 on page 4: It should not be ith here. i should be a total number, or it should be expressed as n. Line 146 on page 5: How many samples were surveyed in the field?

**Response:** Thank you for the specific suggestions. We address those issues in revised manuscript. Recently, we published a 2020 annual report by Global Ecosystems and Environment Observation Analysis Research Cooperation (http://www.chinageoss.org/geoarc/2020/) through cooperation between the Global Earth Observation System of Systems (GEOSS) and the National Remote Sensing Center of China at the Ministry of Science and Technology. We developed a set of new algorithms to retrieve the UIS and UGS fractions using sub-pixel decomposition method through random forest algorithm using Google Earth Engine (GEE) platform. We improved the methods on mapping UIS and UGS fractions and updated the datasets of CLUD-Urban product.

**5 Mapping UIS and UGS fractions using GEE platform**

5.1 Collection of training samples

[revised manuscript text omitted]

**Comment 8:** Line 151 on page 6: What is the higher resolution? Does visual interpretation take the smallest unit of Google images as a single pixel to calculate the number of impervious and vegetation units? Line 153 on page 6: "a" should be deleted. Line 154 on page 6: "densities" would be better to be "fractions". Line 155 on page 6: It should be "values in the same area were". Line 157 on page 6: "shows" should be "showed". Line 157 on page 6: There are two ".". Line 158 on page 6: It should be ", respectively" and "validation of". Line 167 on page 6: It should be a new sentence form "note". Line 169-171 on page 6: How is the urban area defined in this study? Are you using existing data and method? If so, it cannot prove the advantages of this study.

**Response: Response:** Thank you for the specific suggestions. We address those issues.

**Comment 9:** Line 174 on page 6: What is the actual urban expansion rate? Can you give a value to prove the similarity? Line 175 on page 6: "other" here should also be deleted. Line 181 on page 6: Does the UGS here refer to urban green space or areas with high green space fraction? Line 184 on page 7: The "main urban areas" here may not be a very appropriate statement. Line 188 on page 7: Since there are other components, why don't you say high proportional UGS represents parks and greenbelts with ecological functions? Line 197 on page 7: "was" should be "were". Line 199 on page 7: Please be consistent with the previous. Determine to use "dataset" or "datasets" to express the UISA and UGS data?

**Response:** Thank you for your comments. We revised the manuscript according the suggestions.

**Comment 10:** Table 1: "Note" should be left aligned. Table 4: There should be a "Note" before "MRE: : :". Figures: All maps lack a compass. Figure 4: What are the meanings of the small pictures on the right? Please make more explanations. Figure 8: There are two sets of legends in the figure, and some colors are similar. How to distinguish them?

**Response:** We revised the form and notes on Table 1, Table 4 and all figures.

---

## Author Comment (AC2) · 17 Sep 2020

Thank you for the comments and suggestions. These comments were very helpful for revising and improving our paper. We have responded to the comments point by point and made the detailed revisions embedded in the manuscript with the line numbers indicated in the responses.

**Comment 1:** This paper developed a 30-m resolution dataset of China's urban impervious surface area and green space fraction from 2000 to 2018. The fraction information of impervious and green space is revealed from remotely sensed indicators. I have some comments on the adopted approaches and presented results.

Main comments (1) It is difficult to derive the logistic regression model directly since most of the available observations are limited in this study (i.e., only five observations). Why not use continuous observations (i.e., annual) to fit the logistic regression model.

**Response:** Thank you for your comments. Recently, we published a 2020 annual report by Global Ecosystems and Environment Observation Analysis Research Cooperation (http://www.chinageoss.org/geoarc/2020/) through cooperation between the Global Earth Observation System of Systems (GEOSS) and the National Remote Sensing Center of China at the Ministry of Science and Technology. We developed a set of new algorithms to retrieve the UIS and UGS fractions using sub-pixel decomposition method through random forest algorithm using Google Earth Engine (GEE) platform. We improved the methods on mapping UIS and UGS fractions and updated the datasets of CLUD-Urban product.

**Changes in manuscript:** We rewrote the method on the retrieval of UIS and UGS

fractions below in L170-205.

**Comment 2:** Retrieving the ISA information from the NDVI directly seems not reasonable. First, the maximum NDVI used in this study may be fluctuated over years and across spaces. Second, for the built-up areas in arid regions, the proposed approach of estimating the impervious surface information from the NDVI is not reliable. This is an issue that needs to be well addressed.

**Response:** Thank you for your comments. We updated an algorithm to acquire the UISA and UGS fractions. We found the new method and datasets have a high reliability to address those issues.

**Changes in manuscript:** We updated the method on the retrieval of UGS fraction in Line 170-205.

**5 Mapping UIS and UGS fractions using GEE platform**

5.1 Collection of training samples

[revised manuscript text omitted]

**Comment 3:** The comparison of urban areas with other products is necessary, including the accuracy and urban area. It is inadequate if only presenting these comparable figures here directly.

**Response:** Thank you for your comments. Currently, the mainstream datasets on mapping global or China's urban land use/cover focus on the urban boundaries or impervious surface areas. We conducted a comparison on the different datasets and provided on the associated analysis.

**Changes in manuscript:** Here we added a section "7.3 Comparisons of the CLUD-Urban product with other datasets" to compare the accuracy between CLUD-Urban

and the existing dataset in L270-285.

**Comment 4:** A similar approach to estimate the fraction of green space is not reasonable also. I am wondering if this linear relationship can well estimate the green space.

**Response:** Thank you for your comments. In the new dataset, we acquired the UGS fraction through random forest algorithm using Google Earth Engine (GEE) platform. We added the section to elucidate the producing flow on the new version CLUD-Urban dataset.

**Changes in manuscript:** We rewrote the method on the retrieval of UIS and UGS fractions below in L170-205.

**Comment 5:** Minor comments: Page 3, Line 78-80: how to harmonize the spectral bands between Landsat and HJ (or CBERS-1)? Page 3, Line 95: the author mentioned that most classifications exclude the green space, which is not accurate. Green space is a kind of definition from land use, which is consists of trees, shrubs, grasses etc., which are general land cover types. Page 4, Line 100: the definition of new and old urban lands from their colors still needs more evidence. Visually, the new built-up areas such as residential which has similar layout and materials may be similar to the old urban lands Page 8, Line 230: there is a repeated reference (Dong et al., 2017).

**Response:** Thank you for your comments.

**Changes in manuscript:** We revised the manuscript on the data sources and pre-processing in L120-125. We rewrote the sentence in Line 95. The definition of new and old urban lands in Landsat image isn't essential, so we replaced this image by a new image in Fig. 2. We removed the repeated reference in L370.

---

## Author Comment (AC3) · 17 Sep 2020

Thank you for the comments and suggestions. These comments were very helpful for revising and improving our paper. We have responded to the comments point by point and made the detailed revisions embedded in the manuscript with the line numbers indicated in the responses.

**Comment 1:** The authors present multi-year maps of urban imperviousness and greenness of China, which were estimated based on hand-drawn urban boundary and the relationship between vegetation greenness and surface imperviousness. Despite the data might be valuable to a variety of urban-related applications, there are many uncertainties remain. As a data set, these uncertainties should be clearly addressed so that users could better use it. First, using NDVI as the only indicator to estimate surface imperviousness is problematic. The NDVI-based method would overestimate the extent of impervious surfaces because of their similar characteristics as some land uses/covers on NDVI images, especially bare ground. This is especially true in most Chinese cities as they have seen substantial expansions during the study period and the extent of bare ground cannot be ignored. Second, calibration of NDVI-ISA relationship is not clear in many aspects. For example, how was ISA reference measured for model calibration? What was the performance of region averaged model compared to city-specific ones?

**Response:** Thank you for your comments. Recently, we published a 2020 annual report by Global Ecosystems and Environment Observation Analysis Research Cooperation (http://www.chinageoss.org/geoarc/2020/) through cooperation between the Global Earth Observation System of Systems (GEOSS) and the National Remote Sensing Center of China at the Ministry of Science and Technology. We developed a set of new algorithms to retrieve the UIS and UGS fractions using sub-pixel decomposition method through random forest algorithm using Google Earth Engine (GEE) platform. In newly developed CLUD-Urban product, we adopted the advantage of high accuracy and long-time series in mapping urban land from CLUD. We also utilized the highly efficient computation and large storage capacities on GEE platform. In mapping CLUD-Urban product, we proposed to quantitively retrieve the UIS and UGS fractions using random forest. The new CLUD-Urban product exhibits a high accuracy and reliability in delineating urban land surface property. Therefore, we uploaded the new version datasets on national UIS and UGS fractions dataset with 30m resolution in 2000, 2005, 2010, 2015 and 2018.

**Changes in manuscript:** We rewrote the fifth part on "5. Method of mapping UIS and UGS fractions using GEE platform", including three sub-sections: the collection of training samples, retrieval of settlement and vegetation fractions using random forest, and mapping of UIS and UGS fractions in L170-205.

**5 Mapping UIS and UGS fractions using GEE platform**

**5.1 Collection of training samples**

The training samples of UIS and UGS fractions are a pivotal input parameter in random forest model for mapping national settlement and vegetation fraction. In light of large discrepancies among UIS and UGS composites in different climate zones with various geographical and social economic conditions, we collected a total of 2,570 samples from randomly selected cities in different climate zones (Schneider et al. 2010) (Fig. 5). Here we also refer to the existing UIS dataset to acquire samples with 10% intervals of the ISA fraction, and those samples primarily distributed in the homogeneous UIS or UGS areas, which might provide more effective samples and decrease the impact of imagery mismatch. The samples of UIS and UGS covered with diversified types, including buildings, roads and squares, and

grass, trees from parks, road and residential green spaces. The UIS and UGS percentages were interpreted within each sample using Google Earth images (Fig. 5b1-b4). Finally, the training samples in 2000, 2005, 2010, 2015 and 2018 were used for training the random forest model, respectively.

---

## Author Comment (AC5) · 17 Sep 2020

Thank you for the comments and suggestions. These comments were very helpful for revising and improving our paper. We have responded to the comments point by point and made the detailed revisions embedded in the manuscript with the line numbers indicated in the responses.

**Comment 1:** The manuscript provides the China's 30-m UISA and UGS fraction datasets based on the urban area in CLUDs by logistic regression and linear calibration using NDVI from Landsat data. I would suggest the authors reorganize the sections, give more details on the samples and mapping algorithm, discuss more about the accuracy of the maps and comparison with various datasets, add a discussion part and resubmit this paper.

**Response:** Thank you for the comments. We considerably rewrote the sections in Method, Results, and Discussion. In this revision, we proposed a new mapping strategy to acquire the multitemporal and fractional information of the essential urban land cover types at national scale through synergizing the advantage of both big data processing and human interpretation in aid of geoknowledge. We developed a set of new algorithms to acquire the UIS and UGS fractions using random forest algorithm in GEE platform. And then the UIS and UGS fractions with 30 m × 30 m resolution were mapped through overlaying the urban boundaries of CLUD.

Here we added five sections to elucidate the mapping strategy and technological flow on developing the new version CLUD-Urban product, including the strategy of developing CLUD-Urban product, data sources and preprocessing, extraction of urban boundaries from CLUD, method of mapping UIS and UGS fractions using GEE platform, accuracy assessment of the CLUD-Urban product and comparison of different products.

In discussions, "8.1 The mapping advantages integrated with human-computer interpretation and GEE platform; 8.2 The potential implications in promoting habitat environment and sustainability of cities; 8.3 Limitations of the method and dataset and future prospect" were added to address those issues.

**Changes in manuscript:** We conducted a major revision on the method, results and discussions in L105-330.

**Comment 2: Figures:**

 For Figure 5 and 8, it would be better to remove the other land cover types and only show the fraction of UISA. The color is confusing among the vegetation types and the lower percentage of UISA.

**Response:** We revised the legend of Fig. 9.

**Comment 3:** Introduction:**

 There are quite a few existing dataset/report that are providing information about urban green spaces and urban land use categories of China. For examples, (1) https://www.mdpi.com/2072-4292/10/10/1569/htm, (2) https://www.sciencedirect.com/science/article/pii/S2095927319307054?via%3Dih ub

**Response:** We citied the above references, and added the reviews on those researches. **Changes in manuscript:** We added the reference in L385-550

**Comment 4: Method**

3. I feel the methodology section can be written to make it clearer (e.g., sample selection, the retrieval of UGS fraction)

**Response:** Thank you for your suggestions. We rewrote the method parts, including the strategy of developing CLUD-Urban product, data sources and preprocessing, extraction of urban boundaries from CLUD, method of mapping UIS and UGS fractions using GEE platform, accuracy assessment of the CLUD-Urban product and comparison of different products.

**Changes in manuscript:** We rewrote the method on the extraction of urban boundaries from CLUD and the retrieval of UIS and UGS fractions below in L130-205.

**5 Mapping UIS and UGS fractions using GEE platform**

**5.1 Collection of training samples**

The training samples of UIS and UGS fractions are a pivotal input parameter in random forest model for mapping national settlement and vegetation fraction. In light of large discrepancies among UIS and UGS composites in different climate zones with various geographical and social economic conditions, we collected a total of 2,570 samples from

randomly selected cities in different climate zones (Schneider et al. 2010) (Fig. 5). Here we also refer to the existing UIS dataset to acquire samples with 10% intervals of the ISA fraction, and those samples primarily distributed in the homogeneous UIS or UGS areas, which might provide more effective samples and decrease the impact of imagery mismatch. The samples of UIS and UGS covered with diversified types, including buildings, roads and squares, and grass, trees from parks, road and residential green spaces. The UIS and UGS percentages were interpreted within each sample using Google Earth images (Fig. 5b1-b4). Finally, the training samples in 2000, 2005, 2010, 2015 and 2018 were used for training the random forest model, respectively.

---

## Author Response (AR2)

Thank you for the comments and suggestions. These comments are very helpful for revising this paper. We revised this paper and responded to the comments point by point.

**Reviewer #1:**

**Comment 1:** The authors have made a good effort to address previous concerns. But I have a few more concerns about this revision. One of my concerns is the definition of the urban area and the derived urban boundary. How did CLUD define the urban area and distinguish the urban areas from rural settlements and others (industrial and traffic lands)?

**Response:** Thanks for your comments and suggestion. I agreed with you that the separation of urban area from rural settlements and others is difficult if no extra information is used. Fortunately, in our case, we have the CLUD data and define the urban area based on visual interpretation on the Landsat images by support of field survey and satellite images, thus, we can make sure that we produce accurate urban area by excluding rural settlements and other possible areas that may not belong to urban area.

In CLUD dataset, the construction land was divided into three second-level classes – urban land, rural settlements, and others (See Figure 2 and Figure 3). Urban land was defined as a built-up area of the concentrated construction, i.e. buildings, roads, squares, green infrastructure and other lands for providing the living, industrial production, and ecosystem services for the dwellers of cities or towns (Kuang, 2020a). They can be megacities (more than 10 million population), megalopolis (5-10

million population), large cities (1-5 million population), medium cities (0.5-1 million population), small cities (0.2-0.5 million population), and towns (less than 0.2 million population).

In digital interpretation for producing CLUD, we built firstly the detailed image interpretation symbols for each second-level land use class from Landsat or similar resolution images. Usually, the polygons of urban lands exhibit larger sizes than rural settlements and others (industrial and traffic lands) in cinerous color ornamenting with white. The digitalized personnel can differentiate the urban land from rural settlements and others based on the established interpretation symbols and geo-knowledge from field investigation.

**Changes of manuscript:** We added texts to provide the definition of urban area in 4.1 The classification system and interpretation symbols (See L140-145).

We added an explanation on how to distinguish the urban areas from rural settlements and others in section (4.2 Land use and dynamic polygon interpretation) (See L150-160).

In Figure 12, it looks like the core built-up area is classified as an urban area, which also includes industrial and traffic lands. I understand that the vector polygons of urban boundaries were converted to raster data with 30 ✕ 30 m cell size, but how about the uncertainty of vector polygons since the human-computer interaction may induce the errors.

**Response:** Indeed, the industrial and traffic lands located in built-up area are

contained in urban land. However, the industrial and traffic lands outside cities are excluded in the definition of urban land. We provided Figure 3 to show the polygons of both code 51 and code 53.

The urban vector boundaries were acquired from Landsat images or similar resolution images. Therefore, the vector polygons were converted to raster data with 30 m ✕ 30 m resolution. The accuracy of vector polygons from the human-computer interaction were assessed. The users' accuracy of urban land type is relatively high with 93.67% in 2010, 92.65% in 2015, and 91.32% in 2018 (Table 2).

**Changes of manuscript:** We added texts to give explanations in section (4.1 The classification system and interpretation symbols).

We added a sentence in section (4.3 Retrieval of multitemporal urban boundaries) to explain the issue.

This paper aims to track the long-term UGS/UIS change in China therefore the validation of change maps is of great importance. It would be better to give more information on the UGS/UIS validation samples within the changing area every year, e.g. how many changed validation samples every five years? what is the accuracy of the change samples of UIS and UGS?

**Response:** Thank you for your comments. Yes, it is important to produce the urban change image and provide the accuracy of change results. We considered this condition to validate the UIS and UGS fraction utilizing the Google Earth images from corresponding period and acquired the validation samples with more than 30%

of total samples.

Therefore, 1070 validation samples of total 1869 samples were located in changed areas at an interval of three years or five years. The R and RMSE for the changed UIS are in 2000-2018 0.88 and 0.12, and for the changed UGS area in 2000-2018 are 0.88 and 0.12.

**Changes of manuscript:** We added the accuracy assessment for changed UIS and UGS (L250-255). We revised a sentence in section (6 Accuracy assessment of CLUD-urban product).

Figure 1. the text "China's urban impervious surface area and green space fractions in 2000 2018" are enveloped by the cylinder.

**Response:** Thank you for identifying this problem. We modified it.

Figure 11. the captions didn't match the text in the figures (i.e., northeastern, also in captions of Figure 9). China spelled as "Chian". It would be better to give a map showing the different zones in China.

**Response:** Thank you for identifying this problem. The new caption is changed as

Figure 11: The urban impervious surface (UIS) and urban green space (UGS) fractions at national and regional scales (coastal, central, eastern and western zones) in 2000 and 2018.

The wrong spelled word was corrected.

Figure 9: The spatial distribution of urban impervious surface (UIS) in selected cities

from 2000 to 2018. (DEM dataset was downloaded from SRTM 90 m Digital

Elevation Data (http://srtm.csi.cgiar.org/))

**Reviewer #2:**

**Comment 1:** Fundamental revisions have been made, and as a pleasing result, the

manuscript quality has greatly improved.

It is also suggested to use the dots to represent the sampling cities in Figure 5.

However, it is up to the authors.

**Response:** Thank you for your comment. We revised Figure 5 using the dots as the

sampling cities.

---

## Author Response (AR3)

Comments to the Author:

 Thank you for your efforts in improving the manuscript. Please proofread the response and correct the typos such as "The R and RMSE for the changed UIS are in 2000-2018 0.88 and 0.12, and for the changed UGS area in 2000-2018 are 0.88 and 0.12". Moreover, the detail for the evaluation of changed UIS and UGS should be added. Please correct these problems before the paper can be accepted.

**Response:** Thanks for your comments and suggestion. We revised this sentence (Lines 254-255) and added more details for the evaluation of changed UIS and UGS (Line 237, Lines 239-240).